# A maximum-entropy model to predict 3D structural ensembles of chromatin from pairwise distances with applications to interphase chromosomes and structural variants

Guang Shi [1,3] ✉ & D. Thirumalai [1,2] ✉

The principles that govern the organization of genomes, which are needed for an understanding of how chromosomes are packaged and function in eukaryotic cells, could be deciphered if the three-dimensional (3D) structures are known. Recently, single-cell imaging techniques have been developed to determine the 3D coordinates of genomic loci in vivo. Here, we introduce a computational method (Distance Matrix to Ensemble of Structures, DIMES), based on the maximum entropy principle, with experimental pairwise distances between loci as constraints, to generate a unique ensemble of 3D chromatin structures. Using the ensemble of structures, we quantitatively account for the distribution of pairwise distances, three-body co-localization, and higher-order interactions. The DIMES method can be applied to both small and chromosome-scale imaging data to quantify the extent of heterogeneity and fluctuations in the shapes across various length scales. We develop a perturbation method in conjunction with DIMES to predict the changes in 3D structures from structural variations. Our method also reveals quantitative differences between the 3D structures inferred from Hi-C and those measured in imaging experiments. Finally, the physical interpretation of the parameters extracted from DIMES provides insights into the origin of phase separation between euchromatin and heterochromatin domains.

In little over a decade, a variety of experimental techniques, combined with computational tools and physical modeling, have greatly contributed to our understanding of chromatin organization in numerous cell types and species[1–9]. These studies have paved the way toward a deeper understanding of the relationship between genome structure and gene expression[10–14]. The commonly used experimental techniques could be broadly classified as sequence-based or microscopy-based. The former include the Chromosome Conformation Capture (3C)[15] and its variants, including Hi-C[16] and Micro-C[17], which in concert with high-throughput sequencing provide population-averaged data for the contact matrix or contact maps (CMs)[18,19]. The elements of the CM are the average probabilities that

[1]Department of Chemistry, University of Texas at Austin, Austin, Texas 78712, USA. [2]Department of Physics, University of Texas at Austin, Austin, Texas 78712, USA. [3]Present address: Department of Materials Science, University of Illinois, Urbana, Illinois 61801, USA. ✉e-mail: guang.shi.gs@gmail.com; dave.thirumalai@gmail.com

two loci separated by a given genome length ($s$, a linear measure) are in spatial proximity. In order to reveal the cell-to-cell chromatin variations in chromatin conformations, single-cell Hi-C (scHi-C) or similar techniques have also been developed[20–24]. These studies, along with methods that utilize a combination of Hi-C and imaging techniques[25], reveal the statistical and heterogeneous nature of chromatin organization. In addition, methods like SPRITE[26] and genome architecture mapping (GAM)[27,28], which avoid the ligation step in Hi-C[9], have also revealed the organization of chromosomes, including features that are missed in conventional Hi-C methods, such as the higher-order contacts that go beyond pairwise interactions. How to utilize the data contained in the CM to directly reconstruct an ensemble of the three-dimensional structures of the genome is a difficult inverse problem. Data-driven approaches[29–38] have been advanced to solve the complicated Hi-C to 3D structure problem (see the summary in ref. [39] for additional related studies and ref. [40] for a comprehensive overview of the existing methods).

The imaging-based technique is the most direct route for determining the 3D chromosome structures[41–43]. In combination with the fluorescence in situ hybridization (FISH) technique[44], imaging experiments have enabled direct imaging of the position of the genomic loci at the single-cell level. The FISH experiments have revealed global genome organization principles, such as the chromosome territories (CT)[44]. Recently developed multi-scale multiplexed DNA FISH-based imaging methods[25,45–52] have further advanced the field, which has resulted in measurements of the spatial positions of many loci for a large number of cells, thus providing not only glimpses of the structures over a large length scale but also a quantitative assessment of the fluctuations in the cell-to-cell conformations. For instance, the imaging method was used to obtain the locations of ~65 loci in a 2-megabase-pair (Mbp) region for chromosome 21 (Chr21)[47] for a large number of cells. More recently, the method was further improved to image over ~900 targeted loci spread uniformly across the entire chromosome scale (≈242 Mbp for Chr2) and over 1000 genomic loci across all 23 chromosomes[51]. Compartment features, long-range interactions between loci, and the distribution of the radius of gyration could be directly visualized or calculated using the coordinates of the imaged loci, thus providing direct quantitative information on the nuances of genome organization. Although the resolution in the imaging technique will doubtless increase in the future, currently Hi-C based methods provide higher resolution at the CM level but not at the 3D structural level. Thus, by combining the experimental data from a variety of sources and computational methods unexpected insights about chromosome organization could be gleaned.

Here, we introduce a method, DIMES (from DIstance Matrix to Ensemble of Structures), that utilizes the mean distance matrix (DM) between loci as input to generate an ensemble of structures using the maximum entropy principle. The data in the two studies[47,51] are used to quantitatively validate the DIMES method. In order to demonstrate the predictive power of the method, we used DIMES to determine the changes in the organization (expressed as CM) from structural variants (inversion) on Chr1 from the mouse cell line and the effect of single loci deletion on Chr2 from IMR90 cell line. Our approach, when applied to Hi-C (using the HIPPS method[53]) and imaging data, reveals important differences between the two methods in the finer details of the structural ensemble of Chr21.

## Results

### Formulating DIMES using the maximum-entropy model

We developed the DIMES method, which utilizes imaging data, to generate an ensemble of 3D chromosome conformations. The input for our theory is the pairwise distances between the genomic loci (Fig. 1). We seek to find a joint distribution of positions of loci, $P(\{x_i\})$, which is consistent with the squared mean pairwise distance $\langle ||x_i - x_j||^2 \rangle = \langle r^2_{ij,\exp} \rangle$, where $\langle r^2_{ij,\exp} \rangle$ is the experimentally measured average squared distance between two loci $i$ and loci $j$. One could also use the average distance instead of the average squared distance as constraints. However, constraining average squared distances is computationally more efficient because the resulting maximum-entropy distribution is a multivariate Gaussian distribution which allows fast sampling.

In general, there are many, possibly infinite, number of such $P(\{x_i\})$ which satisfy the constraints. Using the maximum entropy principle[54,55], we can find a unique distribution $P^{\mathrm{MaxEnt}}(\{x_i\})$ whose differential entropy is maximal among all possible distributions. We should point out that the maximum entropy principle has been previously used in the context of genome organization[56–58], principally to learn the values of the unknown parameters in a chosen energy function deemed to be appropriate for describing chromosome organization. Recently, it was shown that the maximum-entropy distribution with the constraints of contact frequency can be mapped to a confined lattice polymer model in ref. [59]. The Lagrange multipliers that enforce the constraints are interpreted as the contact energies in the Hamiltonian of the polymer with the position of each monomer occupying the lattice sites. Here, we use the pairwise distances as constraints and derive the corresponding maximum-entropy distribution from which the 3D structures may be readily obtained.

The maximum-entropy distribution $P^{\mathrm{MaxEnt}}(\{x_i\})$ is given by,

$$P^{\mathrm{MaxEnt}}(\{x_i\}) = \frac{1}{Z}\exp\left(-\sum_{i<j} k_{ij}||x_i - x_j||^2\right), \qquad (1)$$

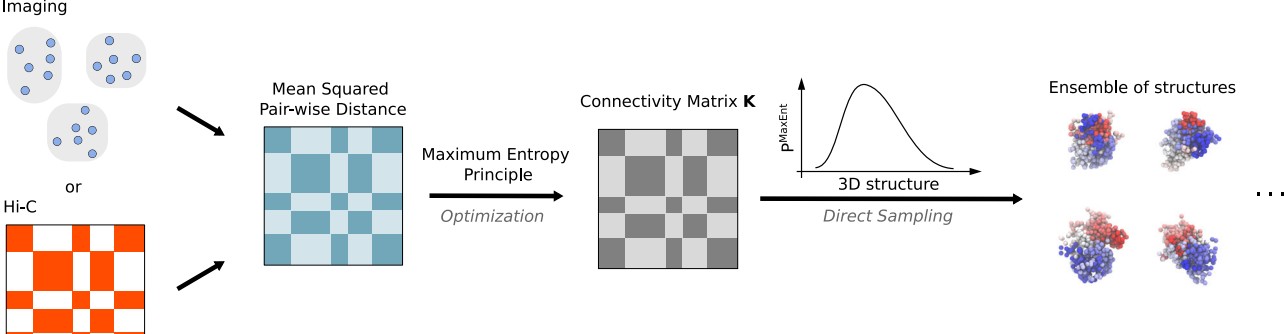

**Fig. 1 | Schematic flowchart for DIMES.** Either imaging (measurements of chromatin loci coordinates) or Hi-C data (contact map) may be used to compute or infer the mean pairwise distance matrix, which is used as constraints to determine the maximum-entropy distribution $P^{\mathrm{MaxEnt}}$. The parameters, which we refer to as the connectivity matrix $K$, in the $P^{\mathrm{MaxEnt}}$, are obtained through an optimization procedure using either iterative scaling or gradient descent algorithm, as explained in the "Methods" and Supplementary Note 1. The ensemble of structures (coordinates of chromatin loci) can be randomly sampled from the distribution $P^{\mathrm{MaxEnt}}$.

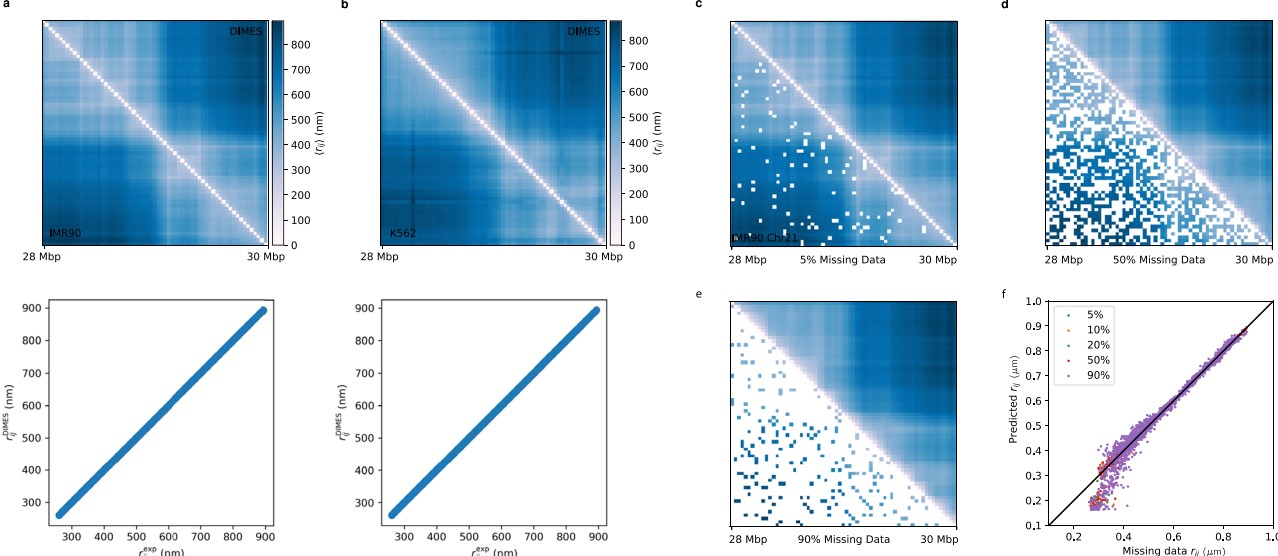

**Fig. 2 | Effectiveness of DIMES in matching the targets generated using experimental imaging data.** Comparison between the mean spatial distances computed from the reconstructed structures and the experimental data in cell lines: IMR90 (**a**), K562 (**b**), A549 (Supplementary Fig. 10a), and HCT116 (Supplementary Fig. 10b). The upper panel shows side-by-side comparisons of the distance matrices, and the lower panel displays the scatter plot between individual pairwise distances $r_{ij}$'s. The Pearson correlation coefficient is near unity (>0.99) for all of the cell types which shows the accuracy of the DIMES method. **c**–**e** Comparison between input distance map with missing data (lower triangle) and the full predicted distance map (upper triangle). A percentage of pairwise distances in the distance map are randomly chosen and set to be missing data (displayed in white). **f** $r_{ij}$ for missing data versus the predicted values. The black line has a slope of unity.

where $Z$ is the normalization factor, and $k_{ij}$'s are the Lagrange multipliers that are determined so that the average value $\langle \|\boldsymbol{x}_i - \boldsymbol{x}_j\|^2 \rangle = \langle r^2_{ij,\,\mathrm{exp}} \rangle$. It can be shown that $P^{\mathrm{MaxEnt}}(\{\boldsymbol{x}_i\})$ is a multivariate normal distribution (see "Methods"). In addition, it can be proven that for any valid $r^2_{ij,\,\mathrm{exp}}$, there exists a unique set of $k_{ij}$ (see "Methods"). The values of $k_{ij}$ can be determined using an iterative scaling algorithm[60] or a gradient descent algorithm (see "Methods" and Supplementary Note 1). For later reference, we define the matrix with elements $k_{ij}$'s as $\boldsymbol{K}$. Note that Eq. (1) has the same form as the Boltzmann distribution of the generalized Rouse model (GRM) with $k_{ij} \geq 0$, which has been applied to reconstruct chromosome structure by fitting to the Hi-C contact map[61,62]. However, it is important to point out that Eq. (1) is derived under the maximum entropy principle, which does not assume the thermal equilibrium condition of the system.

The three steps in the DIMES to generate an ensemble of chromosome structures are: (1) We compute the target mean squared spatial distance matrix from experimental measurements of the coordinates of genomic loci. (2) Using an iterative scaling or gradient descent algorithm, we obtain the values of $k_{ij}$'s to match the experimental measured $\langle r^2_{ij,\,\mathrm{exp}} \rangle$. (3) Using the values of $k_{ij}$, the coordinates of the 3D chromosome structures can be sampled from $P^{\mathrm{MaxEnt}}(\{\boldsymbol{x}_i\})$—a multivariate normal distribution. The details of the procedures are described in "Methods".

## Validating DIMES

In order to demonstrate the effectiveness of DIMES, we first used the experimental data[47] in which the authors reported, using a highly multiplexed super-resolution imaging approach, coordinates of about 65 individual loci for the 2-Mbp segment for Chromosome 21 (Chr21) from four different cell lines (IMR90, K562, HCT116, and A549). Using the calculated mean spatial distance matrices from the reported coordinates as targets, we determined $k_{ij}$ (Eq. (1)), which allowed us to generate an ensemble of structures for this 2-Mbp segment. For all cell lines, the mean spatial distance matrices computed from the reconstructed ensemble of structures almost perfectly match the target distance maps (Fig. 2a, b and Supplementary Fig. 9). In addition, we also applied the DIMES to the chromosome-scale data (see a later section), and achieved the same level of accuracy.

We then perform cross-validation of the DIMES method. This is done as follows: a randomly chosen fraction of pairwise distances from the distance map is deemed to be missing data. The resulting distance map containing the missing data is then used as input for DIMES. It is important to note that the missing data is not used to update $k_{ij}$. Finally, the predicted distances for the missing data are compared with the values obtained from the full distance map. Figure 2c–f compare the input distance maps with missing data and the full predicted distance maps. Remarkably, DIMES quantitatively predicts pairwise distances for the missing data even if only 10% of the distance map is used. Together, these results demonstrate that the model is effective in producing 3D structures that are consistent with the experimental input and is robust with respect to missing data.

## Distribution of pairwise distances

Next, we tested if DIMES could recover the properties of the genome organization that are not encoded in the mean spatial distances. We focused on reproducing the distributions of pairwise distances, which can be calculated because pairwise distance data are available for a large number of individual cells. It is worth emphasizing that the input in the DIMES method is the mean spatial distance, which does not contain any information about the distributions.

To quantitatively measure the degree of agreement between the measured and calculated distance distributions using DIMES, we compute the Jensen–Shannon divergence (JSD), defined as, $(D(p\|m) + D(q\|m))/2$ where $p$ and $q$ are two probability vectors and $D(p\|m)$ is the Kullback–Leibler divergence. The JSD value is bounded between 0 and 1. A zero value means that the two distributions are identical. For each loci pair $(I, j)$, we calculated the JSD, thus generating the JSD matrix. Figure 3a shows the JSD matrix, and Fig. 3b displays the histogram of all the calculated JSDs. The average value of JSDs is merely 0.02, which shows that the overall agreement between the calculated and measured distributions of distances is excellent.

Upon closer inspection of Fig. 3b, we find that the values of JSDs are not randomly distributed. We choose two pairs with relatively large and small JSD values. Comparison between the experimental and

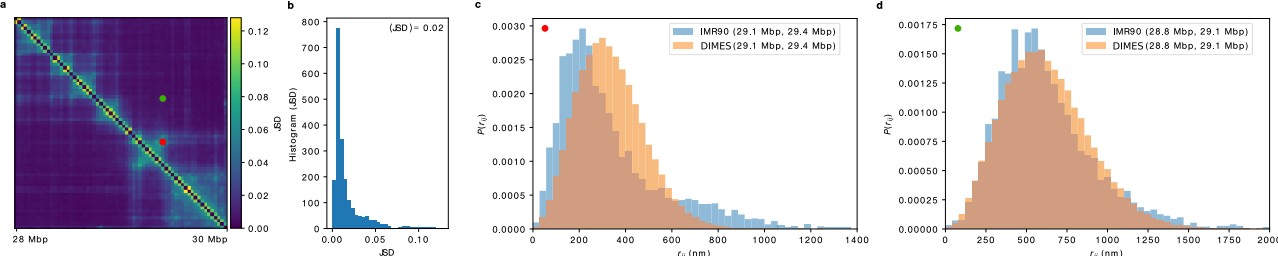

**Fig. 3 | Comparison between the calculated and measured $P(r_{ij})$. a** The Jensen–Shannon divergence (JSD) matrix computed from experiments (top half) and using DIMES (bottom half). The scale on the right shows that the maximum JSD value is ≈only 0.12, thus establishing the effectiveness of DIMES in calculating the $P(r_{ij})$. **b** Histogram of the JSD. **c** Comparison between $P(r_{ij})$ for the pair (29.1 Mbp, 29.4 Mbp) for the experiment and the model (corresponding to the red dot in (**a**)). **d** Same plot as in (**c**) but for the pair (28.8 Mbp, 29.1 Mbp), which corresponds to the green dot in (**a**).

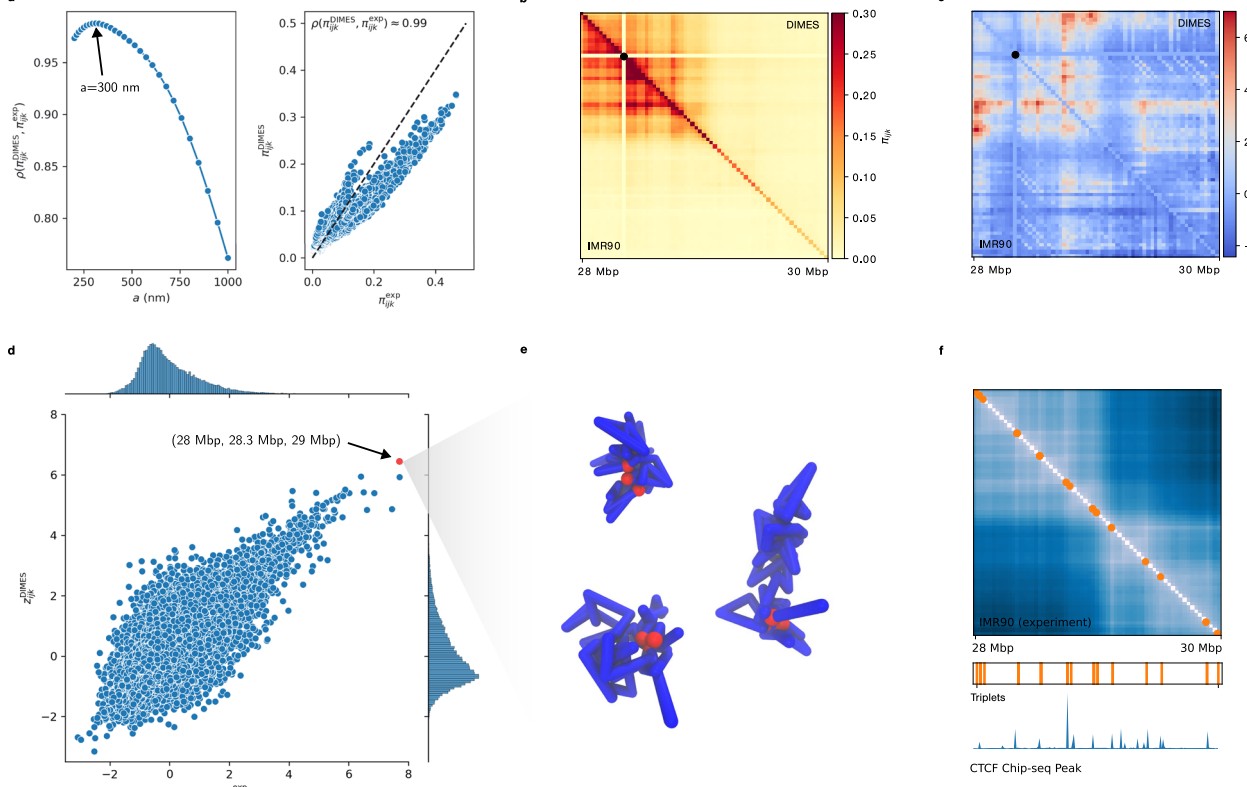

**Fig. 4 | Comparison of three-way contacts predicted by DIMES with imaging experimental data. a** (Left) Pearson correlation coefficient, $\rho(\pi_{ijk}^{\text{exp}}, \pi_{ijk}^{\text{model}})$ versus $a$, where $a$ is the threshold distance for contact formation. $\pi_{ijk}^{\text{exp}}$ ($\pi_{ijk}^{\text{DIMES}}$) are the probabilities of co-localization of three loci $(i, j, k)$, defined as $\pi_{ijk} = \text{Pr}(r_{ij} < a, r_{jk} < a, r_{ik} < a)$. (Right) Plot of $\pi_{ijk}^{\text{DIMES}}$ versus $\pi_{ijk}^{\text{exp}}$ for $a = 300$ nm at which the $\rho(\pi_{ijk}^{\text{exp}}, \pi_{ijk}^{\text{DIMES}})$ is a maximum. **b** Comparison between the heatmaps for $\pi_{ijk}^{\text{exp}}$ (lower triangle) and $\pi_{ijk}^{\text{DIMES}}$ (upper triangle) for $i = 11$. The scale on the right gives $\pi_{11,j,k}$. **c** Same as (**b**) except it compares $Z_{ijk}^{\text{exp}}$ (lower triangle) $Z_{ijk}^{\text{DIMES}}$.

$Z_{ijk} = (\pi_{ijk} - \mu(\pi_{ijk}))/\sigma(\pi_{ijk})$ where $\mu(\pi_{ijk}) = \sum_{m,n,q} \delta(|j - i||k - j| - |m - n||n - q|)\pi_{mnq}/\sum_{m,n,q} \delta(|j - i||k - j| - |m - n||n - q|)$, and $\sigma(\pi_{ijk})$ is the standard deviation. The $Z_{11,j,k}$ scale is on the right. **d** Scatter plot of $Z_{ijk}^{\text{DIMES}}$ versus $Z_{ijk}^{\text{exp}}$ for $i = 11$. **e** Three individual chromosome conformations with the constraint that loci $(1, 11, 31)$ be colocalized $(a = 300$ nm). **f** (Top) Mean distance map for cell-line IMR90 with 10 triplet sets (orange circles) with 10 largest $Z_{ijk}$ values. (Middle) Same as the orange circles in the top panel, but plotted horizontally for easier comparison with the bottom panel. (Bottom) The CTCF Peak track plotted using Chip-seq data[69].

calculated $P(r_{ij})$ for one pair with JSD = 0.08 (Fig. 3c) and with = 0.009 (Fig. 3d) shows that the dispersions are substantial. The pair with JSD = 0.08 (Fig. 3c) samples distances that far exceed the mean value, which implies that there are substantial cell-to-cell variations in the organization of chromosomes[47,51]. The percentage of the subpopulations, associated with different distances, can in principle be inferred by deconvolution of the full distance distribution[63].

## Co-localization of three loci and biological significance

We next asked if the method accounts for higher-order structures, such as three-way contacts, discovered in GAM[27,28] and SPRITE

experiments[26] and imaging experiments[47], and predicted by the theory[64,65]. First, we computed the probability of co-localization of loci triplets, $\pi_{ijk}(a) = \text{Pr}(r_{ij} < a, r_{ik} < a, r_{jk} < a)$ where $a$ is the distance threshold for contact formation ($r_{ij} < a$ implies a contact). To make a quantitative comparison with experiments, we also computed $\pi_{ijk}^{\text{exp}}(a)$ using the experimental data. We then calculated the Pearson correlation coefficient, $\rho$, between $\pi_{ijk}^{\text{exp}}(a)$ and $\pi_{ijk}^{\text{sim}}(a)$. Figure 4a shows that the degree of agreement between experiment and theory is best when $a$ is in the range of 200–400 nm. We chose $a = 300$ nm at which $\rho$ is a maximum. The scatter plot of $\pi_{ijk}^{\text{DIMES}}$ versus $\pi_{ijk}^{\text{exp}}$ (Fig. 4a) is in excellent agreement with $\rho(\pi_{ijk}^{\text{DIMES}}, \pi_{ijk}^{\text{exp}}) \approx 0.99$.

The good agreement is also reflected in the Fig. 4b, which compares the heatmaps $\pi_{ijk}^{exp}$ for $i = 11$ (lower triangle) and the heatmap for $\pi_{ijk}^{DIMES}$ with $i = 11$ (upper triangle). Due to the polymeric nature of the chromatin, the absolute value of $\pi_{ijk}$ may not be instructive because $\pi_{ijk}$ is usually highest when the three loci $i, j, k$ are close along the sequence. To normalize the genomic distance dependence, and capture the significance of the co-localization of triplets, we calculated the $Z$-score for $\pi_{ijk}$ defined as $Z_{ijk} = (\pi_{ijk} - \mu(\pi_{ijk}))/\sigma(\pi_{ijk})$ where $\mu(\pi_{ijk}) = \sum_{m,n,q} \delta(|j - i||k - j| - |m - n||n - q|)\pi_{mnq}/\sum_{m,n,q} \delta(|j - i||k - j| - |m - n||n - q|)$, and $\sigma(\pi_{ijk})$ is the corresponding standard deviation. Positive $Z$-score implies that the corresponding triplet has a greater probability for co-localization with respect to the expected value. As an example, a comparison of the $Z$-scores for $Z_{ijk}^{exp}$ and $Z_{ijk}^{DIMES}$ in Fig. 4c for $i = 11$ shows excellent agreement. The scatter plot in Fig. 4d of $Z_{ijk}^{DIMES}$ versus $Z_{ijk}^{exp}$ shows that the triplet of loci with index (1, 11, 31) has the highest value of $Z_{ijk}$ both in the experimental data and the predictions based on DIMES. Three randomly selected individual conformations with (1, 11, 31), colocalized within distance threshold $a = 300$ nm (Fig. 4e) adopt diverse structures, attesting to the heterogeneity of chromosome organization[25].

To demonstrate the biological significance of the triplet with $Z_{ijk}$ values, we overlay 10 sets of three-way contacts with 10 largest $Z_{ijk}$ values on the mean spatial distance map (Fig. 4f, with orange circles representing the triplet loci). Interestingly, the three-way contacts are localized on the boundaries of the Topologically Associating Domains (TADs)[66,67], which are enriched with CTCF motifs[66,68]. Comparing the location of the triplets and the CTCF peak track (plotted using Chip-seq data[69]) we find that the spatial localization of the triplets is highly correlated with the CTCF peaks. This implies that the CTCF/cohesin complex has a tendency to co-localize in the form of triplets or possibly higher-order multiplets, which is consistent with the recent experimental studies demonstrating the presence of foci and clusters of CTCF and cohesin in cells[70,71].

## Higher-order structures

Because the DIMES method is quantitatively accurate, we could probe higher-order chromatin structures. To this end, we considered three aspects of chromatin organization, which can be calculated directly from the coordinates of the Chr21 loci.

**TAD-like patterns in a single cell.** Imaging experiments[47] show that even in a single cell, domain-like or TAD-like structures may be discerned, even in inactive X chromosomes whose ensemble Hi-C map appears to be featureless[72]. These signatures are manifested as TAD-like patterns in the pairwise distance matrix (see the left panels for two cells in Fig. 4B in ref. [47]). Similar patterns are observed in the individual conformation generated by DIMES as well (Supplementary Fig. 1). We then compute the boundary probabilities[47] of individual genomic loci using experimentally measured structures and the structures generated by DIMES. Figure 5a shows that DIMES captures the profile of boundary probabilities nearly quantitatively. Interestingly, such TAD-like structures are present even in ideal homopolymer structures (Supplementary Fig. 3). We surmise that the intrinsic features of fluctuating polymer conformations contribute to such TAD-like structures. These structures are dynamic in nature because they are a consequence of fluctuations. The specific interactions which distinguish chromosomes from an ideal homopolymer counterpart lead to the statistically preferred distributions of these TAD-like structures rather than being random.

**Size and shapes.** Using the ensemble of Chr21 structures, we wondered if the radius of gyration ($R_g$) and the shape of the genome organization could be accurately calculated. We determined the $R_g$ distribution, $P(R_g)$, and the shape parameter $\kappa^2$[73,74], for the Chr21 in the 28−30 Mbp region. The results (Fig. 5b, and Supplementary Fig. 10)

show that the model achieves excellent agreement with the experiment, both for $P(R_g)$ and $P(\kappa^2)$.

Given the excellent agreement, it is natural to ask whether the DIMES method can capture the size and shape on finer scales. To shed light on this issue, we calculated the distribution of $R_g$ and $\kappa^2$ ($P(R_g; i, j)$ and $P(\kappa^2; i, j)$) for every sub-segment ($i$ loci to $j$ loci) over the 2-Mbp region. In order to assess the accuracy of the predictions, we calculated the JSD between $P^{DIMES}(R_g; i, j)$ and $P^{exp}(R_g; i, j)$, and between $P^{exp}(\kappa^2; i, j)$ and $P^{DIMES}(\kappa^2; i, j)$. On the finer scale, there are deviations between calculations based on DIMES and the experiment. The JSD heatmap (Fig. 5c) for all pairs $i$ and $j$ shows that the deviation is not uniform throughout the 2-Mbp region. We picked two regions that show good agreement (red segment) and poor agreement (blue segment) in Fig. 5c. Direct comparison of $P(R_g)$ and $P(\kappa^2)$ for these two segments is presented in Fig. 5d. For the blue segment, both the predicted $P(R_g)$ and $P(\kappa^2)$ have less dispersion than the experimental data. This suggests that the heterogeneity observed in experiments is even greater than predicted by the DIMES method. For the red segment, the predicted and experimentally measured are in good agreement. Visual inspection of Fig. 5c suggests that the discrepancy is localized mostly in the TAD regions, which might be due to the dynamic nature of these sub-structures.

**Ensemble of structures partition into clusters.** We then compared the overall distributions of the ensemble of calculated structures with experiment data. To do this, we first performed t-SNE to project the coordinates of each conformation onto a two-dimensional manifold using the distance metric, $D_{mn}$,

$$D_{mn} = \sqrt{\frac{1}{N^2} \sum_{i,j} \left( r_{ij}^{(m)} - r_{ij}^{(n)} \right)^2} \qquad (2)$$

where $r_{ij}^{(m)}$ and $r_{ij}^{(n)}$ are the Euclidean distances between the $i^{th}$ and $j^{th}$ loci in conformations $m$ and $n$, respectively. Based on the density of the t-SNE projections (shown as contour lines in Fig. 6), it is easy to identify two peaks, implying that the space of structures partition into two major clusters. The points are then clustered into two major clusters (orange and blue) using Agglomerative Clustering with the Ward linkage[75]. The percentage of cluster #1 (orange) in the experiment is 36%, which is in excellent agreement with the value (34%) predicted by the DIMES method. We display a representative structure with the lowest average distance to all the other members in the same cluster for the two clusters in Fig. 6. Based on our analyses of the experimental results (conformations on the left in Fig. 6), the representative structure belonging to cluster #1 (in blue) is more compact compared to the one in cluster #2 (in orange). The same trend is found in the DIMES predictions (conformations on the right in Fig. 6).

We also computed the mean distance maps from all the conformations from each cluster (shown on the left and right side in Fig. 6). The distance maps show that the structures in cluster #2 adopt a dumbbell shape whereas those belonging to cluster #1 exhibit no such characteristic. The quantitative agreement between the distance maps from the experiment and the model is excellent.

## Structures of the 242-Mbp-long Chr2

We extend the DIMES method to chromosome-scale imaging data in order to compare with the recent super-resolution imaging experiments, which reported coordinates of 935 loci genomic segments with each locus being 50-kbp long spanning the entire 242-Mbp Chr2 of Human IMR90 cell[51]. Note that there are spaces between the loci that are not imaged. We computed the average squared distance matrix from the measured coordinates and then used DIMES to generate an ensemble of structures. Figure 7a compares the experimental and calculated mean distance matrices using the DIMES method (Fig. 7a).

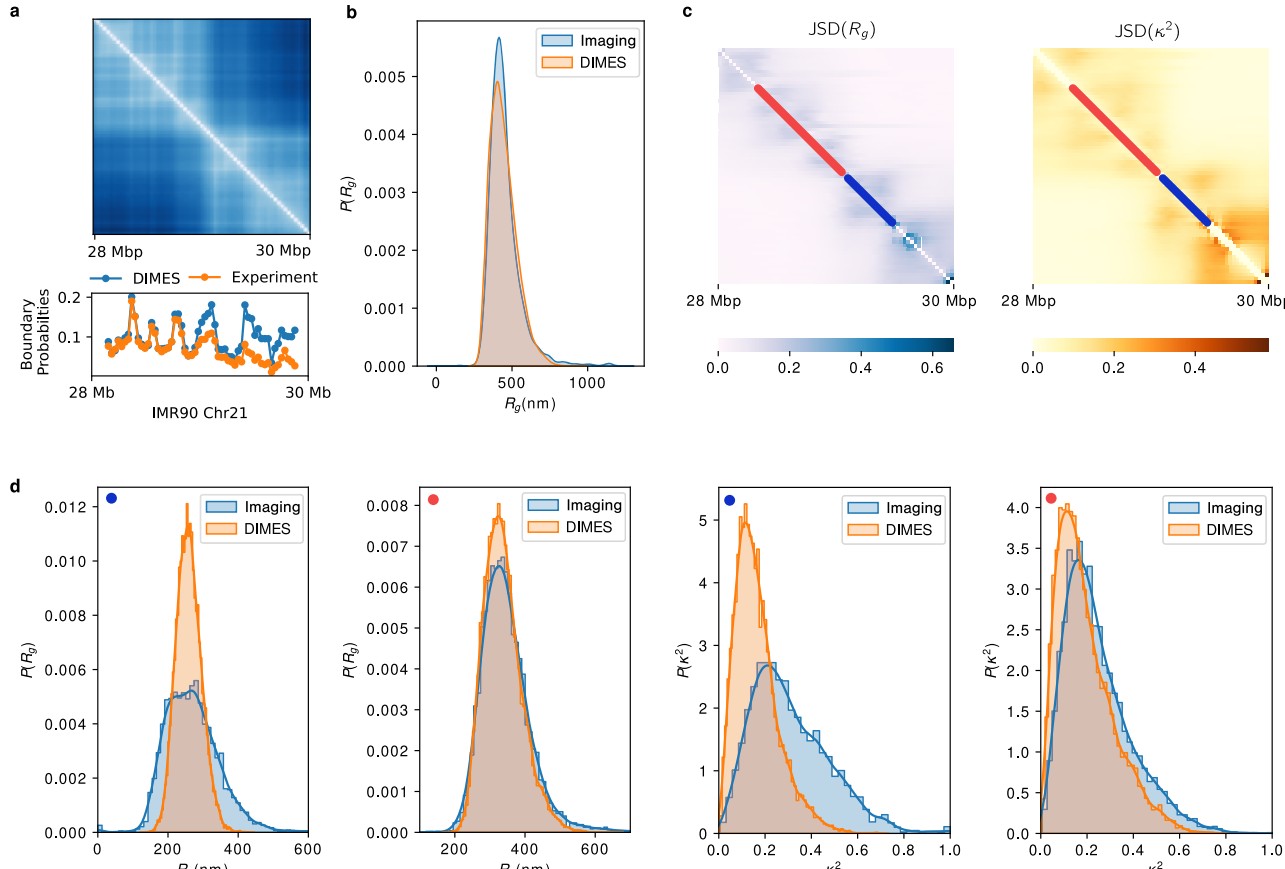

**Fig. 5 | TAD-like structures and shape characteristics of Chromosome 21.**
**a** Comparison of boundary probabilities between experimental imaging data and prediction from DIMES. Boundary probability measures the probability that a genomic locus acts as a single-cell domain boundary. **b** Comparison of $P(R_g)$ between the experiment and the DIMES predictions. $P(R_g)$ is the probability density distribution of the radius of gyration $R_g$ for the 28–30 Mbp region of Chr21. Comparison of $P(\kappa^2)$ where $\kappa^2$ is the shape parameter is shown in Supplementary

Fig. 10. **c** The heatmaps of the JSD of the distribution of $R_g$ and $\kappa^2$ between the experiment and the calculations. Each element $(i, j)$ in the heatmap is the value of JSD for the segment that starts from $i^{th}$ loci and ends at $j^{th}$ loci. Red and blue lines represent two such segments. **d** Comparison of $P(R_g)$ and $P(\kappa^2)$ between the predictions using DIMES and those calculated using experiments for the segments marked in (**c**). The blue (red) dot on the left corner of each sub-figure indicates the locations of the segments.

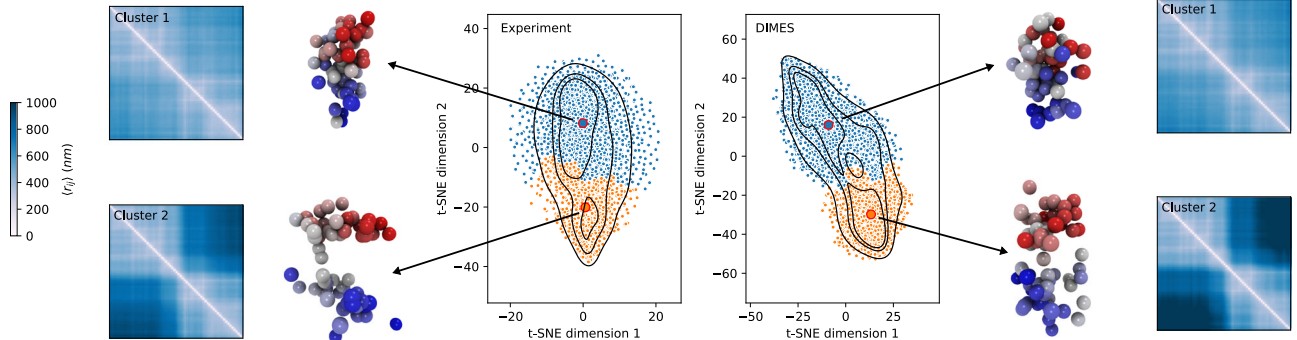

**Fig. 6 | Phase space structure of the 2-Mbp Chr21 organization.** t-SNE projections are calculated from the conformations and the agglomerative clustering results. Individual conformation is projected onto a two-dimensional manifold using t-SNE. Contour lines of the density of t-SNE projections are shown to reflect the underlying clusters of the conformations. The conformations naturally partition into two clusters (cluster #1 and cluster # 2 marked by blue and orange, respectively). A representative conformation from each cluster and the mean distance map computed from the conformations belonging to each cluster are also displayed.

As before, the agreement is excellent (see also Fig. 2). A few randomly chosen conformations from the ensemble are shown in Supplementary Fig. 2a, demonstrating that there are large variations among the structures. Experimental single-cell distance maps (Supplementary Fig. 8a) show that similar variations are also observed in vivo. DIMES reproduces the distributions of pairwise distances at large length scales (Supplementary Fig. 8b).

In addition to recovering the experimental data, our approach produces genomic distance ($s$) dependent effective interaction strengths between the loci, which gives insights into the organization of Chr2 on genomic length scale. Because the parameters in the DIMES are $k_{ij}$'s, which could be interpreted as effective "interaction" strengths between the loci, we asked if the **K** matrix encodes for the A/B compartments (the prominent checker-board pattern in Hi-C experiments

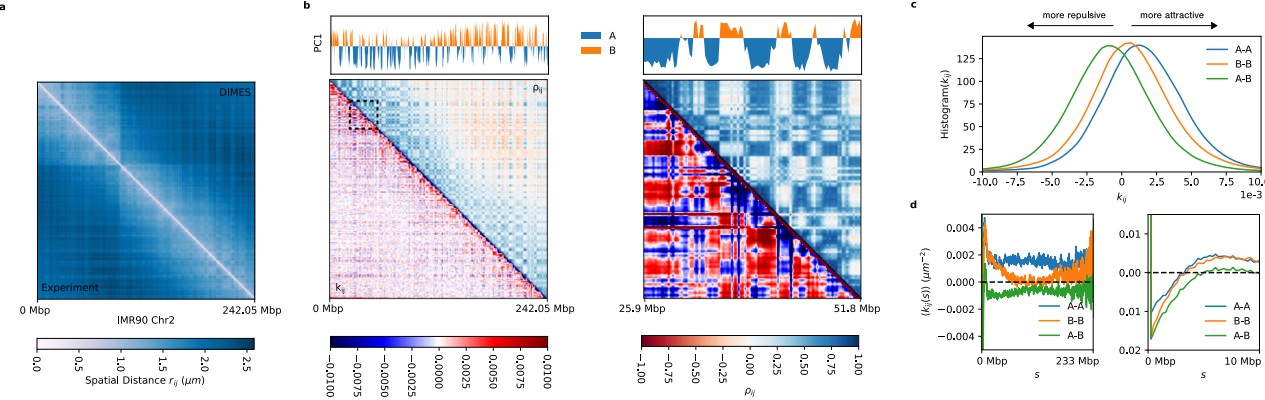

**Fig. 7 | Features of the Chr2 organization. a** The average distance matrix between the experiment (lower triangle) and the DIMES (upper triangle) shows excellent agreement. **b** The connectivity matrix $\mathbf{K}$, whose elements are $k_{ij}$ (lower triangle), and the correlation matrix $\boldsymbol{\rho}$ (upper triangle) computed from $\mathbf{K}$. The top track shows the principal component dimension 1 (PC1) computed using principal component analysis (PCA) from $\boldsymbol{\rho}$. A(B) compartments correspond to negative (positive) PC1. **c** Histogram of $k_{ij}$ for A-A, B-B, and A-B. **d** Genomic-distance normalized $\langle k_{ij}(s) \rangle = (1/(N-s)) \sum_{i<j}^{N} \delta(s-(j-i)) k_{ij}$ for A-A, B-B, and A-B. $\langle k_{ij}(s) \rangle$ are shown for $s$ between 0 and 233 Mbp (left) and for between 0 and 10 Mbp (right).

indicating phase separation between euchromatin and heterochromatin) observed in the distance matrix. (Although $k_{ij}$s may not represent the actual strength associated with interactions between $i$ and $j$, we use this terminology for purposes of discussion.) Note that $k_{ij}$ can be either negative or positive, with a negative (positive) value representing the effective repulsion (attraction). The lower triangle in Fig. 7b shows the matrix $\mathbf{K}$. We then computed the correlation matrix $\boldsymbol{\rho}$ from $k_{ij}$ (see Supplementary Note 2 for details), which is shown in the upper triangle in Fig. 7b. The corresponding principal component dimension 1 (PC1) of $\boldsymbol{\rho}$ is shown in the top panel in Fig. 7b. The negative (positive) PC1 corresponds to A (B) compartments. The results in Fig. 7b show that the A/B compartments can be inferred directly from $\mathbf{K}$, indicating that the parameters in DIMES correctly capture the underlying characteristics of the genome organization on all genomic length scales.

Given that the A/B type of each genomic loci are unambiguously identified, we then computed the histogram of $k_{ij}$ and genomic-distance normalized $\langle k_{ij}(s) \rangle = (1/(N-s)) \sum_{i<j}^{N} \delta(s-(j-i)) k_{ij}$ for A-A, B-B, and A-B interactions. The results show that the mean A-B interactions are repulsive (negative $\langle k_{ij}(s) \rangle$) whereas A-A and B-B interactions are attractive (positive) (Fig. 7c). This finding explains the compartment features observed in the Hi-C data and the distance maps. Furthermore, we find that, on an average, A-A interactions are more attractive than B-B interactions (Fig. 7c), which seems counterintuitive because of the general lore that heterochromatin (formed by B locus) appears to be denser than euchromatin (composed of locus A) in microscopy experiments[41]. On the other hand, the same analysis on Chromosome 21 shows opposite results, with the B-B interactions being stronger on an average than A-A interactions (Supplementary Fig. 4b). The genomic-distance normalized $\langle k(s) \rangle$ also shows that over the range of 2–10 Mbp, $\langle k(s) \rangle$ for B-B is consistently stronger than that for A-A (Supplementary Fig. 4c, d). The results for Chr2 and Chr21 show that the comparison between the strength of A-A and B-B interactions is possibly chromosome-dependent. As we noted in a previous study[8], what is important is that the Flory $\chi = (\epsilon_{AA} + \epsilon_{BB} - 2\epsilon_{AB})/2$ is positive to ensure compartment formation. Here, $\epsilon_{AA}$, $\epsilon_{BB}$, and $\epsilon_{AB}$, are the interaction energy scales involving A and B.

The genomic-distance normalized $\langle k_{ij}(s) \rangle$ for Chr2 shows that all the interaction pairs have the highest value at $s = 1$–a manifestation of the polymeric nature of chromatin fiber. Beyond $s = 1$, all pairs have negative $k(s)$, indicating repulsive interaction on a small length scale. At $s \approx 5$ Mbp, $k(s)$ for all pairs develop positive peaks. At length scale $s > 10$ Mbp, the B-B interactions decay as $s$ increases whereas

$s$-dependent A-A interactions fluctuate around a positive value. In addition, the histograms of $k_{ij}$ (Fig. 7c) suggest that the average differences among A-A, B-B, and A-B interactions are small ($\langle k_{AA} \rangle = 0.0014$, $\langle k_{BB} \rangle = 0.0005$, $\langle k_{AB} \rangle = -0.0009$), which is consistent with the recent liquid Hi-C experiment[76]. In summary, the results in Fig. 7 demonstrate that the application of DIMES to large-length-scale imaging data explains the origin of compartments on large length scales. More importantly, the calculated values of $k_{ij}$ provide insights into interactions between the genomic loci on all length scales, which cannot be inferred solely from experiments. Surprisingly, but in accord with experiments[76], the differences in the strengths of interaction between the distinct loci are relatively small.

Besides demonstrating the efficacy of DIMES in determining the ensemble of structures accurately, the calculated $k_{ij}$s explain micro phase segregation of active (A) and inactive (B) loci. It is a surprise that phase separation between A and B emerges from the calculated $k_{ij}$s, without a polymer model with an assumed energy function. Because $k_{ij}$s can be calculated using either the HIPPS[53] or the DIMES method, the differences in compartment formation (segregation between A and B loci) in various chromosomes can be quantitatively inferred.

## Applications

**Impact of genomic rearrangement on 3D organization.** It is known that the 3D organization of chromosomes can change substantially upon genomic rearrangements, such as duplication (increases the length of the genome), deletion (decreases the genome length), or inversion (shuffling of genome sequence while preserving the length). Both deletion and duplication are drastic genomic changes that require recomputing the $\mathbf{K}$ using either contact[53] or distance maps. In contrast, the inversion is a gentler perturbation, and hence the changes in chromosome folding compared to the wild type (WT) can be calculated by treating it as perturbations applied to the $\mathbf{K}$ for the WT.

Once the WT $\mathbf{K}$ is calculated, a perturbation method could be used to determine the variations in the ensemble of genome structures. In particular, we asked whether the 3D structural changes upon rearrangement in the genomic sequence could be predicted by accounting for the corresponding changes in $k_{ij}$. For instance, inversion (Fig. 8a) would correspond to an inversion on the $\mathbf{K}$ (see Supplementary Note 3). For illustration purposes, we applied our method to the experimental Hi-C maps for the WT and a variant with an inversion[77]. To apply DIMES to the two constructs, we first converted the contact probability to the mean spatial distance using the scaling relation $\langle r_{ij} \rangle = \Lambda p_{ij}^{-1/\alpha}$ where we take $\alpha = 4$[53]. Figure 8a shows excellent

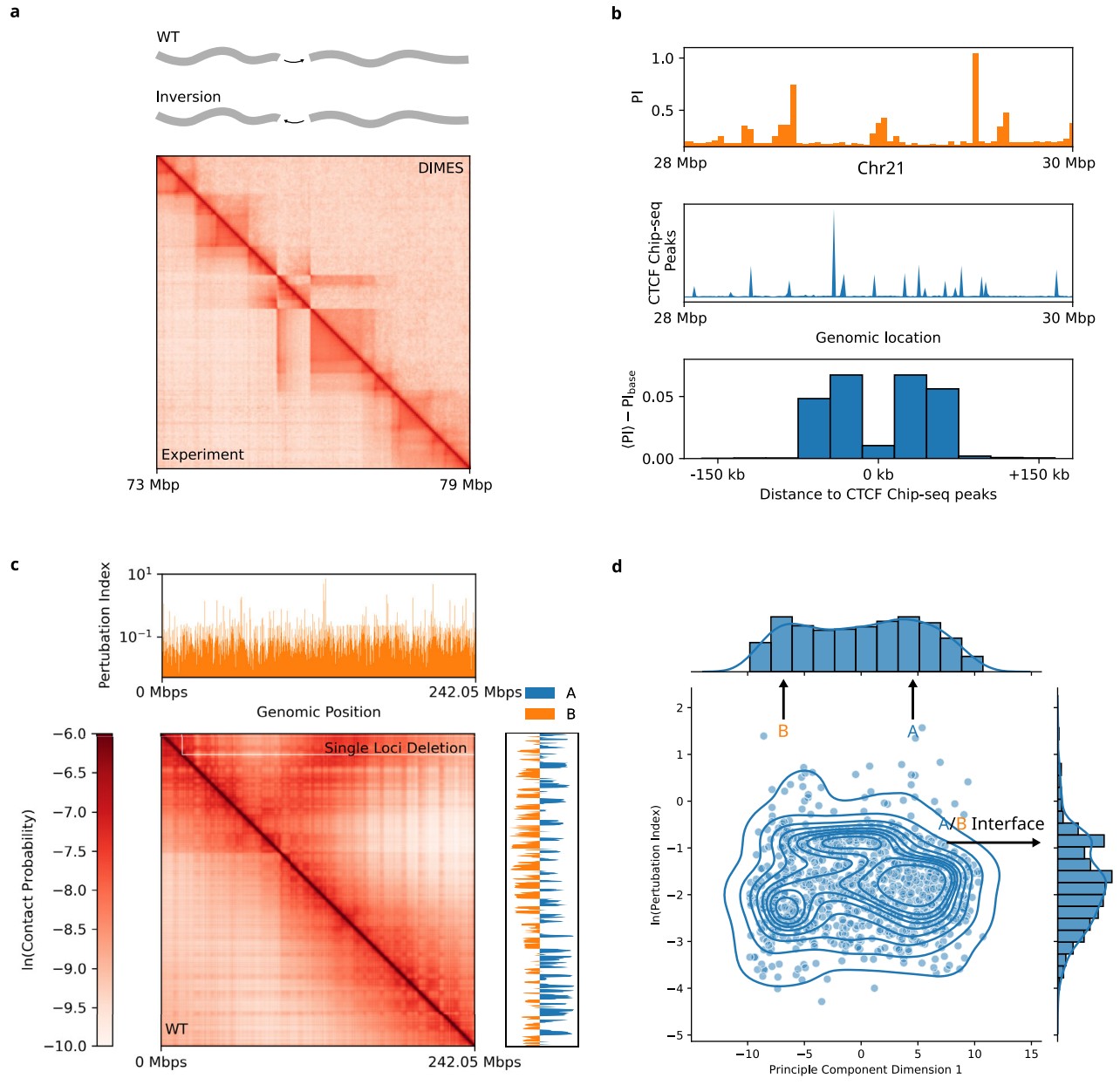

**Fig. 8 | Predictions for structural variants. a** Experimentally measured Hi-C contact map for the 1.1-Mb homozygous inversion for Chr1 from the mouse cell line E11.5 (lower triangle). The position of the segment that is inverted is shown on the track in the top panel. The predicted contact map using DIMES with the perturbation on the WT Hi-C data (see main text and Supplementary Note 3) is shown for comparison (upper triangle). **b** Top: Perturbation Index (PI) for Chr21 28–30 Mbp. Middle: Chip-seq data for CTCF. Bottom: average value of PI as a function of genomic distance from CTCF Chip-seq peaks. Average PI values up to 150 kbp on either side of CTCF Chip-seq peaks are calculated at 30 kbp resolution. $PI_{base} = 0.2$. **c** An example of the contact map with a single locus deletion (upper triangle) for Chr2 from the IMR90 cell line. The top panel shows a track plot of the perturbation index (PI) computed using Eq. (3). **d** Plot of principal component dimension 1 (computed from $\rho$) versus the logarithm of the perturbation index. A, B, and A/B boundaries are marked. Histograms of PC1 and the logarithm of the perturbation index are shown on the top and side, respectively.

agreement between the experimentally measured Hi-C contact map with inversion and the one predicted by the combined HIPPS-DIMES approach. The more drastic structural variants (deletion and insertion) require applying the HIPPS or the DIMES method directly to the mutated CMs or DMs.

**Structural integrity is determined by loci at the CTCF anchors and A/B boundary.** Does the deletion of every single locus have the same effect on the 3D structures? To answer this question, we first investigated the effect of deletion of CTCF/cohesin anchors by applying DIMES to Chr21 28–30 Mbp region[47]. The method for deletion of a single locus is described in Supplementary Note 3. The 3D structural changes are quantified using the Perturbation Index (PI),

$$PI = \sqrt{\left(\sum_{i<j}^{N}(\langle \widetilde{r}_{ij}\rangle - \langle r_{ij}\rangle)^2\right)/\sum_{i<j}^{N}\langle \widetilde{r}_{ij}\rangle^2} \quad (3)$$

where $\widetilde{r}_{ij}$s is the WT average distance between loci $i$ and $j$ and $\langle r_{ij}\rangle$s are changed values after locus deletion. PI profile along the 2-Mbp region and the Chip-seq data for CTCF are shown in Fig. 8b top and middle panels. More importantly, on an average, the PI profile exhibits higher values closer to the CTCF Chip-seq peaks (Fig. 8b

bottom panel), demonstrating that the genomic loci associated with CTCF anchors have a more significant effect on the 3D structures upon their deletion.

We then probe the prediction of locus deletion for Chr2 imaging data[51] (Fig. 8c). The PI profile along Chr2 (Fig. 8c top panel) shows that there are large variations among the individual loci, suggesting that deletion of some genomic locus have a larger impact on the 3D structures than others. To ascertain whether the variations in the PI are associated with the known chromosomal structural features such as A/B compartments, we compared the values of PI with the principal component dimension 1 (PC1) which are computed from correlation matrix $\rho$ (Fig. 8d). The loci with PC1 close to zero are interpreted as the A/B boundaries. We find that, statistically, the boundary elements between A/B compartments have higher PI values, indicated by the basin near $\ln(\text{PI}) = -1$, whereas elements inside the A/B compartments have lower PIs $\ln(\text{PI}) \approx -2$. From this finding, we propose that boundary loci are most important in maintaining chromosome structural integrity.

**Comparing Hi-C and imaging experiment.** Although there are several scHi-C experiments, the majority of the studies report Hi-C data as ensemble averaged contact maps. On the other hand, super-resolution imaging experiments directly measure the coordinates of loci for each cell. It is unclear how the chromosome structures inferred from Hi-C differ from the ones directly measured in imaging experiments. We compare the Hi-C data[1] and the imaging data[51] for Chr2 from the IMR90 cell line. Both the HIPPS and the DIMES methods first convert the contact probability $p_{ij}$ to mean spatial distance $\langle r_{ij} \rangle$ using $\langle r_{ij} \rangle = \Lambda p_{ij}^{-1/\alpha}$. We determined the value of $\Lambda \approx 0.36 \mu m$ and $\alpha \approx 5.26$ by minimizing the error between the distances inferred from Hi-C and the experimental measurements from imaging experiments, $\chi = (2/N(N-1)) \sum_{i<j}^{N} (\langle r_{ij}^{\text{Hi-C}} \rangle - \langle r_{ij}^{\text{Imaging}} \rangle)^2$. Figure 9a compares the mean distance matrix inferred from Hi-C using the HIPPS method and the one computed using the coordinates directly measured in the imaging experiment. Visual inspection suggests that the distance matrix inferred from Hi-C shows stronger compartmental patterns compared to the imaging result even though on an average the mean pairwise distances $r_{ij}$'s obtained from the two methods are in good agreement with each other (Fig. 9b). We also find that the locations of A/B compartments obtained from both the experimental methods are in excellent agreement (Fig. 9c).

Although the positions of the A/B compartments obtained from both Hi-C and imaging data agree with each other, it is unclear whether these two methods could be used to generate 3D structures that are consistent with each other. To ascertain the 3D structures inferred from Hi-C data and the imaging experiments are consistent with each other, we calculated $Q_k$ and $F_k$[53]; $Q_k$ measures the degree of spatial mixing between A and B compartments.

$$Q_k = \frac{1}{N} \sum_i |n_A(i;k)/\tilde{n}_A - n_B(i;k)/\tilde{n}_B|, \quad (4)$$

where $k$ is the number of the nearest neighbors of loci $i$. In Eq. (4), $n_A(i;k)$ and $n_B(i;k)$ are the number of neighboring loci belonging to A compartment and B compartment for loci $i$ out of $k$ nearest neighbors, respectively $[n_A(i;k) + n_B(i;k) = k]$. With $N = N_A + N_B$, the expected number of $k$ neighboring loci in the A compartment with random mixing is $\tilde{n}_A = kN_A/N$ and $\tilde{n}_B = kN_B/N$ where $N_A$ and $N_B$ are the total number of A and B loci, respectively. With $k \ll N$, perfect mixing would result in $Q_k = 0$, and $Q_k \neq 0$ indicates demixing between the A and B compartments.

The function, $F_k$, not unrelated to contact order, quantifies the multi-body long-range interactions of the chromosome structure. We

define $F_k$ as,

$$F_k = \frac{1}{kNF_{0,k}} \sum_i \sum_{j \in m_i(k)} |j - i| \quad (5)$$

where $k$ again is the number of nearest neighbors and $m_i(k)$ is the set of loci that are $k$ nearest neighbors of loci $i$; $F_{0,k} = (1/2)(1 + k/2)$ is the value of $F_k$ for a straight chain. Eq. (5) implies that the presence of long-range interaction increases the value of $F_k$. In both $Q_k$ and $F_k$, $k$ is the number of nearest neighbors for a given locus.

Figure 9d shows that, compared to the results obtained through imaging, the Hi-C method overestimates the extent of long-range interactions, and underestimates the spatial mixing between A and B compartments, which is reflected in the shift of the distribution of $P(Q_k)$ and $P(F_k)$ (we chose $k = 8$ without loss of generality). We also computed the interactions profiles of A-A, B-B, and A-B in the same fashion as shown in Fig. 7c, d from the Hi-C data. The Hi-C data suggest that the B-B interactions are more attractive than A-A, which is the opposite of the results obtained from the imaging data. Furthermore, the extracted $\langle k(s) \rangle$ for $s$ between 0 and 10 Mbp differs in the two methods (Figs. 9f and 7d). The Hi-C data suggest that the interactions within the range of $s$ between 0 and 10 Mbp are attractive (Fig. 9f) whereas the imaging data suggest that interactions for $s \lesssim 5$ Mbp are repulsive, and become attractive for larger $s$ values (Fig. 7d).

Next, we compared the Hi-C and imaging techniques at a smaller scale. To this end, we applied HIPPS/DIMES to Hi-C data for Chr21 28–30 Mbp from the IMR90 cell line[1]. Supplementary Fig. 6a shows the comparison between the mean distance matrix inferred from Hi-C using HIPPS/DIMES method and that calculated from imaging data. The scatter plot between $r_{ij}^{\text{Imaging}}$ and $r_{ij}^{\text{Hi-C}}$ (Supplementary Fig. 6b) shows a higher degree of agreement compared to Chr2 (Fig. 9b). We then compute $F_k$ and its distribution $P(F_k)$, which shows good agreement between Hi-C and imaging data (Supplementary Fig. 6c). These results show that structures inferred from Hi-C and imaging have a higher degree of agreement on the length scale of Mbp compared to the scale of the whole chromosome.

## Discussion

We have developed a computational method (DIMES) that solves the following inverse problem: how to generate the three-dimensional conformations from the experimentally measured average distance matrix? First, applications to genome data on a length scale of a few TADs and on the scale of the whole chromosome, show that DIMES correctly reproduces the pairwise distances. Second, we demonstrate that DIMES accurately accounts for the higher-order structures beyond pairwise contacts, such as the three-body interactions, radius of gyration, shapes, and the clustering of structures. These results for Chr21 on 2-Mbp and the entire 242-Mbp Chr2 agree quantitatively with multiplexed super-resolution data, thus setting the stage for a wide range of applications. Third, we also demonstrate that the DIMES accurately predicts the changes in the structures due to structural variants. We believe that this is a key prediction because the results for the wild type suffice to predict 3D structures, thus eliminating the need to do additional experiments.

### Implications of the DIMES method

Our method is based on the maximum entropy principle, which is used to find the optimal distribution over the coordinates of chromatin loci that are consistent with experimental data. With the choice of the average squared pairwise distances as constraints, the maximum-entropy distribution has a special mathematical structure. The distribution in Eq. (1) shows that: (1) It is a multivariate normal distribution whose properties are analytically known. Thus, finding the values of

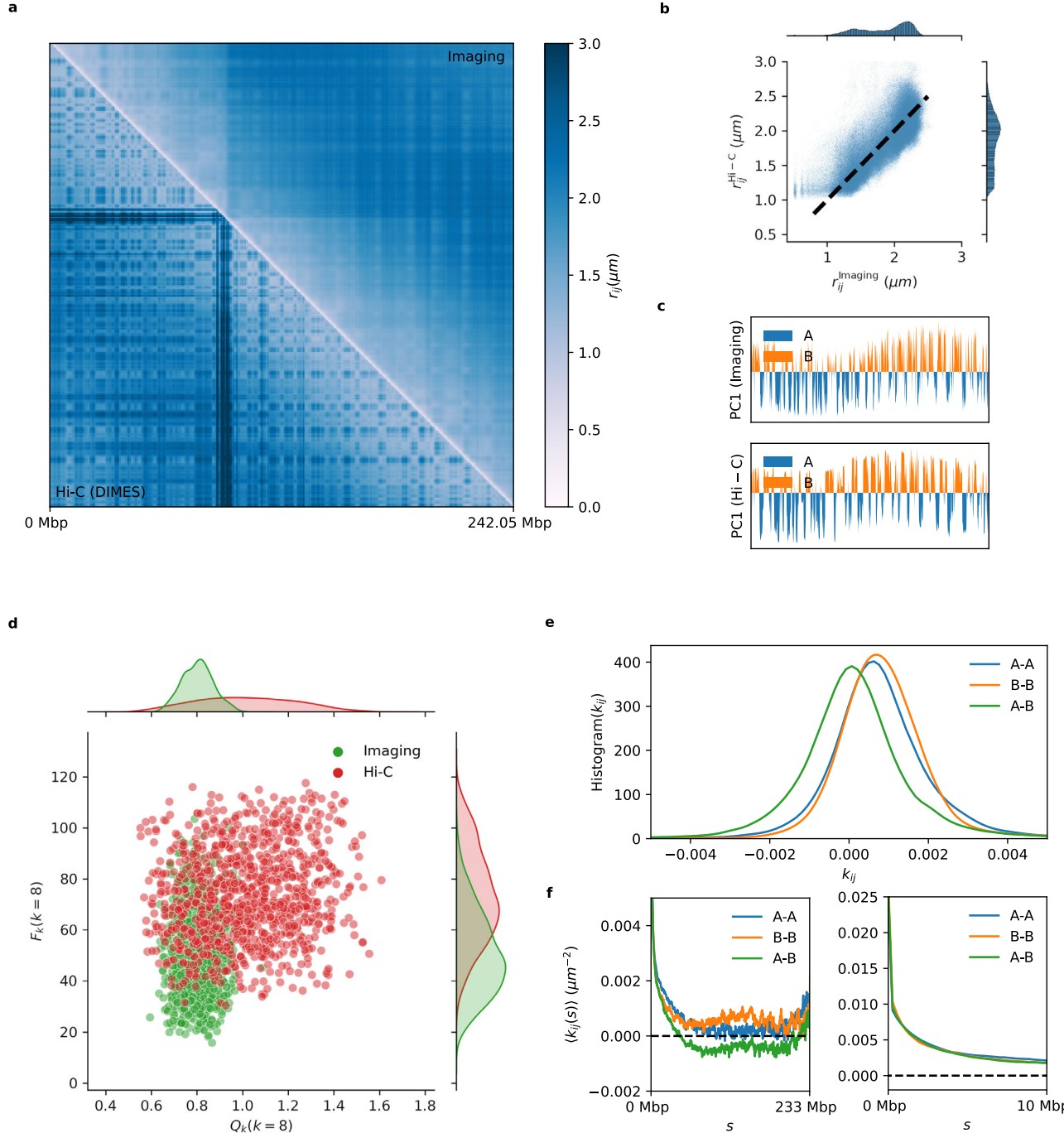

**Fig. 9 | Structural organization calculated from Hi-C and imaging data.**
**a** Comparison between the mean distance matrix inferred from Hi-C contact map (lower triangle) and the experimental measured average distance matrix (upper triangle) for Chr2 from the cell line IMR90. The distance scale is given on the right. **b** Direct comparison of pairwise distances, $\langle r_{ij}^{\text{Imaging}} \rangle$ versus $\langle r_{ij}^{\text{Hi-C}} \rangle$. Each dot represents a pair $(i,j)$. Dashed line, with a slope of unity, is a guide to the eye. **c** Principal component dimension 1 (PC1) for imaging and Hi-C data. The correlation matrix is computed from the connectivity matrix $\mathbf{K}$, and then PCA is performed on the resulting correlation matrix. **d** Scatter plot of $Q_k(k=8)$ and $F_k(k=8)$ for 1000 conformations. The conformations are randomly chosen from the total of ~3000 conformations measured in the imaging experiment (green). For Hi-C, 1000 conformations are randomly generated using HIPPS/DIMES (red). **e** Histogram of $k_{ij}$ for A-A, B-B, and A-B. $k_{ij}$s are obtained using Hi-C contact map. **f** Genomic-distance normalized $\langle k_{ij}(s) \rangle = (1/(N-s)) \sum_{i<j}^{N} \delta(s-(j-i))k_{ij}$ for A-A, B-B, and A-B. $\langle k_{ij}(s) \rangle$ are shown for $s$ between 0 and 233 Mbp (left) and for between 0 and 10 Mbp (right).

parameters $k_{ij}$ does not require simulations but only an optimization procedure. (2) It has the same mathematical structure as GRM[78,79] if one sets $1/k_BT = 1$. Hence, all the properties of the GRM[63] also hold for Eq. (1). This analogy provides a physical interpretation of $k_{ij}$, allowing us to explain phase separation between A and B compartments without appealing to polymer models.

## Interpretation of $k_{ij}$'s

We wondered whether $k_{ij}$'s could be decomposed into two additive terms representing the polymer component and epigenetic component, $k_{ij \in \alpha\beta} = k_0(i,j) + k_e(\alpha,\beta)$; $k_0(i,j)$ is the polymer contribution to $k_{ij}$ which only depends on $i$ and $j$. $k_e(\alpha,\beta)$ is the epigenetic contribution to $k_{ij}$ that only depends on epigenetic types, e.g., $k_e(A,A)$, $k_e(B,B)$, and

$k_e(A, B)$. Assuming that $k_{ij\in\alpha\beta} = k_0(i,j) + k_e(\alpha, \beta)$ holds, by averaging over $s = |i - j|$, it follows that $\langle k_{AA}(s)\rangle - \langle k_{AB}(s)\rangle$, $\langle k_{AA}(s)\rangle - \langle k_{BB}(s)\rangle$, and $\langle k_{BB}(s)\rangle - \langle k_{AB}(s)\rangle$ are independent of $s$. Supplementary Fig. 7 shows that these quantities fluctuate approximately around constant values for $60\,\text{Mbp} < s < 200\,\text{Mbp}$. This result suggests that, to a first approximation, the parameters $k_{ij}$'s may be decoupled into contributions from polymer connectivity and that arising from epigenetic states in an additive manner.

## Predictive power of DIMES

One could legitimately wonder about the utility of DIMES, especially if future imaging techniques generate the coordinates of individual loci at high resolution. Of course, if this were to occur then it would make all computational approaches as well as Hi-C experiments for genome organization irrelevant. However, what is worth noting is that by using the DIMES method one can also predict the 3D structures of structural variants accurately. These applications show that the DIMES method accurately accounts for the data generated by high-resolution imaging experiments for the WT. With the WT $k_{ij}$'s at hand, certain mutational effects could be predicted without having to repeat the imaging experiments, which may not become routine for the foreseeable future. Such high-throughput calculations, which can be performed using DIMES, would be particularly useful when analyzing cancer data from different tissues. We believe this is the major advantage of our computational approach.

## Methods

### Maximum-entropy distribution

In this section, we first show that

$$P^{\text{MaxEnt}}(\{\boldsymbol{x}_i\}) = \frac{1}{Z}\exp\left(-\sum_{i<j}^{N} k_{ij}||\boldsymbol{x}_i - \boldsymbol{x}_j||^2\right), \tag{6}$$

which is the starting point in the DIMES method (see the main text), is a multivariate normal distribution. In the above equation, $\boldsymbol{x}_i = [x_{1i}, x_{2i}, x_{3i}]$ are the 3D coordinates of the $i^{th}$ locus. We write Eq. (6) in a matrix form,

$$P^{\text{MaxEnt}}(\{\boldsymbol{x}_i\}) = \frac{1}{Z}\prod_p \exp\left(\boldsymbol{X}_p^T \boldsymbol{K}\boldsymbol{X}_p\right), \tag{7}$$

where $\boldsymbol{X}_p = [x_{p1}, x_{p2}, \cdots, x_{pN}]^T$, subscript $p \in (1, 2, 3)$ denotes the three spatial dimensions, and $N$ is the total number of loci. The connectivity matrix, $\boldsymbol{K}$, is given by,

$$K_{ij} = k_{ij}, \text{ if } i \neq j \text{ and } K_{ii} = -\sum_{j\neq i} k_{ij}. \tag{8}$$

Eq. (7) is a product of three multivariate normal distributions with covariance matrix $\boldsymbol{\Sigma} = -\boldsymbol{K}^+$ and mean 0, i.e., $\boldsymbol{X}_p \sim \mathcal{N}(\boldsymbol{0}, \boldsymbol{\Sigma})$ for $p = 1, 2, 3$ ($\mathcal{N}(\boldsymbol{0}, \boldsymbol{\Sigma})$ is the multivariate Gaussian distribution). $\boldsymbol{K}^+$ is the Moore-Penrose inverse (pseudoinverse) of $\boldsymbol{K}$. Note that Moore-Penrose inverse ensures that, even if $\boldsymbol{\Sigma}$ is not full rank, the distribution is properly defined. In DIMES, $\boldsymbol{\Sigma}$ has exact one zero eigenvalue, which corresponds to the zero mode (center of the system).

Note that for $P^{\text{MaxEnt}}(\{\boldsymbol{x}_i\})$ to be a normalizable probability density distribution, it is known that $\boldsymbol{\Sigma}$ has to be positive semidefinite ($\boldsymbol{\Sigma} \geq 0$). $\boldsymbol{\Sigma}$ with zero eigenvalue (not full rank) corresponds to the degenerate case, which is expected for a system that is translationally invariant. Hence, it is necessary that $\boldsymbol{K}$ be negative semidefinite. If $k_{ij} \geq 0$ for all $i, j$, as is the case for the Generalized Rouse Model[63,78], $\boldsymbol{K}$ can be proven to be negative semidefinite. Even if $k_{ij} < 0$ for some $i, j$, $\boldsymbol{K}$ can still be negative semidefinite. This means that negative $k_{ij}$ values are allowed as long as the matrix $\boldsymbol{K}$ remains semidefinite.

Next, we prove that there exists a unique set of $k_{ij}$s that satisfy the set of constraints, which in our problem is specified the average squared pairwise distances, $\langle ||\boldsymbol{x}_i - \boldsymbol{x}_j||^2\rangle = \langle r_{ij}^2\rangle$. The constraints could be measured or calculated.

The distribution of coordinates of the loci is given by Eq. (6). Following the derivation in our previous work[53,63], we have,

$$\langle r_{ij}^2\rangle = 3\omega_{ij}^2 \tag{9}$$

where $\omega_{ij}^2 = \Sigma_{ii} + \Sigma_{jj} - 2\Sigma_{ij}$, and $\boldsymbol{\Sigma} = -\boldsymbol{K}^+$. Note that the whole set of equations $\omega_{ij}^2 = \Sigma_{ii} + \Sigma_{jj} - 2\Sigma_{ij}$ has the same number of unknown variables ($\Sigma_{ij}$) as the number of equations, which leads to a unique solution for $\Sigma_{ij}$ given the values of $\langle r_{ij}^2\rangle$. We may obtain $\boldsymbol{K}$ in principle using $\boldsymbol{K} = -\boldsymbol{\Sigma}^+$. Because $\boldsymbol{\Sigma}$ is unique, so is $\boldsymbol{K}$.

Note that this also shows that $\boldsymbol{K}$ can be obtained directly without any optimization procedure. However, there are three issues associated with this method. First, this involves a matrix inversion operation, which is usually numerically unstable for a large matrix. Second, if the target pairwise distance matrix is not a proper distance matrix (for example it does not satisfy triangle inequality) then the resulting $\boldsymbol{\Sigma}$ is not positive semidefinite, which would result in an invalid distribution (Eq. (7)). Third, the direct calculation does not allow for regularization (see Supplementary Note 1), which is needed in practice.

### Calculating the mean squared pairwise distance matrix from experiment data

If the input data is direct measurements of the 3D coordinates of certain loci in the chromatin, then the mean squared pairwise distances between all pairs are computed as,

$$\langle r_{ij}^2\rangle = \frac{1}{M}\sum_{m=1}^{m=M} ||\boldsymbol{x}_i^{(m)} - \boldsymbol{x}_j^{(m)}||^2 \tag{10}$$

where $\boldsymbol{x}_i^{(i)}$ and $\boldsymbol{x}_j^{(i)}$ are the 3D coordinates of $i$ and $j$ loci in the $m^{th}$ single cell and $M$ is the total number of cells for which measurements are made.

If the input data is the Hi-C contact frequencies, then the mean squared pairwise distances between all the pairs are computed using[53],

$$\langle r_{ij}^2\rangle = \left(\Lambda p_{ij}^{-1/\alpha}\right)^2 \tag{11}$$

where $\Lambda$ sets the length scale and $\alpha$ determines the power-law relation between spatial distances and contact probability. In practice, we choose $\alpha = 4$ if it is not determined experimentally. The value of $\Lambda$ can only be determined when the length scale of the system is known. For instance, if the average radius of the gyration of the system is known, $\Lambda$ could be determined by matching the average radius of the gyration of the chromosome from the model with the known value. In practice, we simply set $\Lambda = 1$ if not explicitly specified. Note that the structures obtained with $\Lambda = 1$ can be simply rescaled.

### Optimization

Let us denote the target mean squared pairwise distances as $\langle r_{ij,\,\text{exp}}^2\rangle \equiv a_{ij}$. The updating scheme for the values of $k_{ij}$ in the iterative scaling method[80] is,

$$k_{ij}(t+1) = k_{ij}(t) + \frac{\gamma}{\sum_{i<j}\langle r_{ij}^2\rangle(t)}\ln\frac{\langle r_{ij}^2\rangle(t)}{a_{ij}} \tag{12}$$

$t$ is the step number, and $\langle r_{ij}^2\rangle(t)$ is the value of mean squared distance between loci $i$ and $j$ with parameters $k_{ij}(t)$ at step $t$. The value of the constant learning rate, $\gamma$, is chosen to be $\gamma = 10$ because it gives good convergence speed while ensuring that the result converges.

For gradient descent (GD), the updating scheme for $k_{ij}$ is,

$$k_{ij}(t+1) = k_{ij}(t) - \gamma \left[ \langle r_{ij}^2 \rangle(t) - a_{ij} \right] \qquad (13)$$

where $\gamma$ is the constant learning rate.

For both of these two methods, the value of $\langle r_{ij}^2 \rangle(t)$ needs to be evaluated. In DIMES, the fact that the maximum-entropy distribution is a multivariate normal distribution allows us to compute $\langle r_{ij}^2 \rangle(t)$ directly using the Eq. (9). With values of $k_{ij}(t)$, the matrix $\mathbf{K}(t)$ is constructed according to Eq. (8). Then, the matrix $\mathbf{\Sigma}(t)$ is calculated using $\mathbf{\Sigma}(t) = -\mathbf{K}^+(t)$ where $\mathbf{K}^+(t)$ is the Moore-Penrose inverse of $\mathbf{K}(t)$. Finally, $\langle r_{ij}^2 \rangle(t)$ is computed using $\langle r_{ij}^2 \rangle(t) = 3\left[ \Sigma_{ii}(t) + \Sigma_{jj}(t) - 2\Sigma_{ij}(t) \right]$.

Once the values of $\langle r_{ij}^2 \rangle(t)$ are obtained, the values of $k_{ij}$ are updated using Eq. (12) for iterative scaling or Eq. (13) for GD.

## Generation of structures

After $T$ number of iteration steps, a reasonably converged $\mathbf{K}(T)$ is obtained. Denote $\mathbf{K} \equiv \mathbf{K}(T)$. To generate an ensemble of structures, we sample the distribution given by Eq. (7). Following the derivation in our previous work[53,63], the coordinates of loci, $\mathbf{X}_p$ ($p$ denotes the three spatial dimensions), can be computed as a linear combination of normal modes, $\mathbf{X}_p = \mathbf{V}^T \mathbf{R}_p$ where $\mathbf{R}_p$ are the normal modes. $\mathbf{V}$ is obtained by eigendecomposition of $\mathbf{K}$, i.e., $\mathbf{V}\mathbf{K}\mathbf{V}^T = \mathbf{\Omega} = \mathrm{diag}(\omega_1, \omega_2, \cdots, \omega_N)$. Each component of the normal modes, $R_{i,p}$, is a Gaussian random variable with distribution $\mathcal{N}(0, -\omega_i^{-1})$. Note that different $R_{i,p}$ are independent of each other.

The procedures used to generate an ensemble of structures involve the following steps:

(a)  Perform eigendecomposition of $\mathbf{K}$. Obtain $\mathbf{V}$ and the eigenvalues $\omega_1, \omega_2, \cdots, \omega_N$.

(b)  Draw a total of $N$ random numbers from distributions $\mathcal{N}(0, -\omega_i^{-1})$ for $i = 1, 2, \cdots, N$. Denote these random numbers as $R_1, R_2, \cdots, R_N$.

(c)  Define $\mathbf{R} = [R_1, R_2, \cdots, R_N]^T$. Compute $\mathbf{X}$ using $\mathbf{X} = \mathbf{V}^T \mathbf{R}$.

(d)  Repeat (b) and (c) two more times, each for a spatial dimension. Finally, we obtain $\mathbf{X}_1, \mathbf{X}_2, \mathbf{X}_3$, representing the $x$, $y$, and $z$ coordinates of $N$ number of loci.

(e)  Repeat (b) - (d) $M$ times, resulting in a total number of $M$ randomly sampled 3D structures.

## Reporting summary

Further information on research design is available in the Nature Portfolio Reporting Summary linked to this article.

## Data availability

The data that support this study are available from the corresponding authors upon reasonable request. The Multiplexed FISH imaging data used in this study are publicly available from the GitHub repository at https://github.com/BogdanBintu/ChromatinImaging and Zenodo repository[81] at https://zenodo.org/record/3928890#.Yizd1xDMKFF. The Hi-C data used in this study are publicly available from the GEO database under accession numbers [GSE92294] and [GSE63525].

## Code availability

The code for the DIMES method presented in this work and its detailed user instruction can be accessed at the GitHub repository https://github.com/anyuzx/HIPPS-DIMES[82]. The data analysis is performed using Python 3.9 in Jupyter Lab. The Python packages used in data analysis are Scipy, Numpy, and Pandas.

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

## Acknowledgements

We thank Davin Jeong and Sucheol Shin for several pertinent comments on the work. This work was supported by a grant from the National Science Foundation (CHE 19-00033, D. T.) and the Welch Foundation through the Collie-Welch Chair (F-0019, D. T.).

## Author contributions

G.S. and D.T. designed research; G.S. and D.T. performed research; G.S. and D.T. analyzed data; G.S. and D.T. wrote the paper.

## Competing interests

The authors declare no competing interests.
