## [Peer Review File · Nature Communications]

A maximum-entropy model to predict 3D structural ensembles of chromatin from pairwise distances with applications to Interphase Chromosomes and Structural VariantsREVIEWER COMMENTS

Reviewer #1 (Remarks to the Author):

The authors present a simple, yet remarkably accurate computational model, called DIMES, for generating individual chromatin traces based on population-averaged data. The chromatin modeling literature features many chromatin models (primarily based on bulk Hi-C measurements) and it remains unclear how accurate these models are in reproducing some of the single-cell statistical features. Importantly, the authors use single-cell imaging data to address this problem by comparing the predictions of their model with many of the structural features observed in the imaging data. Thus they provide an overall thorough characterization of the strengths and limitations of their model. Furthermore, the authors highlight an exciting application of their model in predicting the structural changes arising upon genetic deletions/inversions of different loci.

The manuscript, despite being very technical, is well presented and easy to read. However, I have a few comments and concerns that should be addressed prior to publication:

Comment 1: The three parts of the paper: A. Establishment of the model and its recapitulation of single-cell structural features. (80% of the manuscript) B. Application of the model to predicting the structural changes upon genetic deletion/inversion perturbations (10% of the manuscript) and C. Extension of the model to Hi-C population average data (10% of the manuscript) seem disproportionate. Parts B and C should be more expanded and part A should be simplified.

Comment 2: Related to comment 1 part C. The modeling based on Hi-C data (Figure 9) was based primarily on the larger scale Hi-C/imaging data of chr2. It is unclear if the differences the authors highlighted still arise on the smaller scale chr21 - 2Mb data. Naively I would assume that inferences/modeling based on Hi-C is more accurate at the smaller distance scale. The smaller scale should be investigated in Figure 9.

Comment 3: It would be really interesting to see how well the model predicts the disappearance of various CTCF/cohesin loops upon deleting specific CTCF anchors. So please extend the prediction results of Figure 8 to the smaller scale chr21 - 2Mb data. This could have important applications for deciding what enhancers should be targeted with genetic tools to affect specific transcriptional inactivation.

Comment 4: Figure 3 highlights a specific potential limitation of the model that perhaps the author can expand on. It appears that the model is worst at recapitulating the distribution of single-cell distances of CTCF "loops". Interestingly, this distribution is less gaussian in the imaging data (with a heavy tail). Would it be possible to expand the model to a mixture model (with 2 gaussians per pair of loci) in which the heavy tail is better captured? Biologically I would assume there are at least 2 states: CTCF or other factors can either be present or absent at the locus in each single-cell configuration.

Comment 5: Figure 4 c is intriguing. However, I am wondering if the author's model is hiding a limitation regarding triplet interactions. Consider 3 loci: $i > j > k$. If locus i is in contact with locus j (i.e. distance $< 200\text{nm}$) is locus j more likely to be in contact with locus k ? I assume that the authors' model predicts independence between these two events? However, imaging data and multi-contact sequencing data show hub-like interactions. Can the authors calculate $p(j \text{ contact } k \mid i \text{ contact } j)$ compared to $p(j \text{ contact } k \mid i \text{ not contact } j)$ for both model and imaging data?

Comment 6: The TAD-like pattern section (page 3) seems less thoroughly treated than the other sections. I would suggest either removal or adding in more thorough statistical analysis.

Comment 7: Can the authors perhaps better separate the biological/physical meaning of K_{ij} - the essential parameters of the model. For instance K_{ij} currently captures both the polymeric nature of chromatin as well as its epigenetic state. Can, for instance, K_{ij} be broken down (i.e. for the chr2 data)

into two components: one polymeric component and one epigenetic A/B component i.e.: $K_{ij} \sim |i-j|^c * K_{A/B}$ where c is a scale factor constant and K_A and K_B are two constants independent of i and j ?

Reviewer #2 (Remarks to the Author):

This paper reports a maximum entropy-based method (DIMES) to build chromosome structure from distances between chromosome loci measure by imaging techniques. The coefficients (k_{ij}) can be determined by an iterative scaling method. An interesting finding is that the contact triplets in the reconstructed models correlate with CTCF binding sites. Analyzing the TAD structures and two chromosome compartments of the reconstructed models is useful. Comparing the distance maps derived from Hi-C data with the model built from image data is novel and valuable. Studying how loci deletion may change the structure of chromosomes is also interesting. However, there are some issues that need to be addressed.

Specific comments:

(1) The review of the literature is very incomplete. Many methods have been developed for reconstructing genome/chromosome structures from contact maps or distance maps derived from Hi-C data. But few have been reviewed.

(2) The maximum entropy approach has some similarity with the maximum likelihood approach for reconstructing chromosome structures from Hi-C data that is based on Gaussian distribution on the distances between chromosome loci in the literature. But the connection is not drawn in the paper.

(3) In the results section, the validation of the method is based on how well the reconstructed model fits all the experimental distance data, which is not sufficiently rigorous and can easily overfit the data. A more rigorous cross-validation procedure should be used. That is, use a portion of distance data to build the chromosome model and then use the remaining distance data to validate the model.

(4) There is no comparison between DIMES and existing distance based methods of reconstructing 3D chromosome structures from Hi-C data that can also be used to reconstruct 3D chromosome structures from image data in this work.

Reviewer #3 (Remarks to the Author):

The authors report a new computational method for modeling ensembles of 3D chromatin structure that relies on pairwise distances between genomic loci derived from direct imaging experiments. The authors specifically apply this new method, DIMES, to results recently reported by Bintu et al (Science 2018, super resolution of 2 Mb region on chr21) and Su et al (Cell 2020, massive multiplex FISH over 23 chromosomes).

Structural ensembles predicted by DIMES, using the maximum entropy principle, are randomly sampled, deriving predicted spatial distances the authors then compare with target distance map measurements. The authors first validate the accuracy of the models predicted by DIMES by comparison with the experimental input. Can the authors elaborate or further clarify the significance of their reported validation results? For example, the authors claim that using squared distances yield better numerical convergence, but I cannot find a comparison of alternative models tested (e.g. average distance).

In this report, the authors use DIMES to make a number of relevant observations related to

chromosome organization, predicting the distribution of pair-wise distances in structural ensembles, and recovering higher-order structures ranging from three-way contacts and TAD-like patterns to the overall size and shape of chromosome organization. When applied to chromosome-scale imaging data (Su et al.), the authors again recover higher-order chromatin structures, including A/B compartments, and further demonstrate significant variation among DIMES ensemble structures for Chr2. Can the authors comment on the specific experimental data that supports this level of variation (noted in Supp Fig 2), as well as elaborate on the level to which gaps between imaging loci would potentially contribute to these observations?

Beyond the development and applications of DIMES, the results presented offer some valuable insight into genome organization. Ensemble structures for chromosome-scale distances suggest that the levels of A-A and B-B compartment genomic distance preferences (inferred as interaction strengths) are different between chromosome 2 and chromosome 21, as well as between differing length scales. This is an interesting finding, and I'm wondering if the authors can speculate on potential mechanisms.

The authors further demonstrate the value of DIMES by applying their methods to examples of genomic structural variants (structures predicted by DIMES are highly consistent with experimental observations). The authors conclude with a comparison of Hi-C and imaging data and identify a number of interesting divergences between these approaches, specifically in the extent of long-range interactions and A-A and B-B compartment interaction strengths.

The developed method and reported findings are of value to the field and worthy of publication. In general, the manuscript would be improved with extended discussion of these findings.

Responses to Reviewers:

Reviewer #1 (Remarks to the Author):

The authors present a simple, yet remarkably accurate computational model, called DIMES, for generating individual chromatin traces based on population-averaged data. The chromatin modeling literature features many chromatin models (primarily based on bulk Hi-C measurements) and it remains unclear how accurate these models are in reproducing some of the single-cell statistical features. Importantly, the authors use single-cell imaging data to address this problem by comparing the predictions of their model with many of the structural features observed in the imaging data. Thus they provide an overall thorough characterization of the strengths and limitations of their model. Furthermore, the authors highlight an exciting application of their model in predicting the structural changes arising upon genetic deletions/inversions of different loci.

The manuscript, despite being very technical, is well presented and easy to read. However, I have a few comments and concerns that should be addressed prior to publication:

Response:

The reviewer has succinctly summarized the major findings of the paper. We appreciate that the referee finds our work novel. In what follows, we address the issues that she or he has raised as fully as we could. We hope the referee is satisfied with the revised version, which has benefitted greatly from the queries posed by the reviewer.

Comment 1: The three parts of the paper: A. Establishment of the model and its recapitulation of single-cell structural features. (80% of the manuscript) B. Application of the model to predicting the structural changes upon genetic deletion/inversion perturbations (10% of the manuscript) and C. Extension of the model to Hi-C population average data (10% of the manuscript) seem disproportionate. Parts B and C should be more expanded and part A should be simplified.

Response:

The proposal by the reviewer is interesting, and is different from the way we viewed our study. Our rationale for writing the paper the way we did, is that the primary goal of the present manuscript is to validate the theory by faithfully reproducing the excellent experimental imaging data. Parts B and C, as referred to by the reviewer, are under one section "Applications". We believe that part A taking more space in the present manuscript than B/C is appropriate, from our perspective.

Nevertheless, we have expanded parts B and C by addressing some of the comments and revising the paper accordingly. We hope the referee finds the changes sufficient. For details, please see the responses to other comments below.

Comment 2: Related to comment 1 part C. The modeling based on Hi-C data (Figure 9) was based primarily on the larger scale Hi-C/imaging data of chr2. It is unclear if the differences the authors highlighted still arise on the smaller scale chr21 - 2Mb data. Naively I would assume that inferences/modeling based on Hi-C is more accurate at the smaller distance scale. The smaller scale should be investigated in Figure 9.

Response:

We thank the referee for the excellent suggestion, which we ought to have considered. We applied our method to the Chr21 - 2Mb Hi-C data and compared the difference between the Hi-C and imaging data. The results are shown in Figure 9 in the revised manuscript as a new figure. The referee is correct that inferences from Hi-C at the small length scale are more accurate, as compared to the imaging data. The reason is that the resolution on short length scale is better in the current experiments. Please see the added text on page 7 in the revised main text and Supplementary Figure 6.

Comment 3: It would be really interesting to see how well the model predicts the disappearance of various CTCF/cohesin loops upon deleting specific CTCF anchors. So please extend the prediction results of Figure 8 to the smaller scale chr21 - 2Mb data. This could have important applications for deciding what enhancers should be targeted with genetic tools to affect specific transactional inactivation.

Response:

We would like to thank the referee for this valuable suggestion. We fully agree with the referee that investigating the deletion of CTCF/cohesin loops is very interesting and should be investigated. We have performed the loci deletion calculation for Chr21 - 2Mb data from Bintu et al 2018. The results are shown in the revised Figure 8b. In addition, the text on page 6 for the section "Structural integrity is determined by loci at CTCF anchors and A/B boundary" is revised accordingly. Figure 8 caption has been updated.

Comment 4: Figure 3 highlights a specific potential limitation of the model that perhaps the author can expand on. It appears that the model is worst at recapitulating the distribution of single-cell distances of CTCF "loops". Interestingly, this distribution is less gaussian in the imaging data (with a heavy tail). Would it be possible to expand the model to a mixture model (with 2 gaussians per pair of loci) in which the heavy tail is better captured? Biologically I would assume there are at least 2 states: CTCF or other factors can either be present or absent at the locus in each single-cell configuration.

Response:

We thank the referee for raising this interesting point. We agree with the referee that the reduced ability of our model to predict the distribution of pairwise at CTCF anchors is due to the “mixture” of the subpopulations. We would like to point out that our previous work (Shi, Guang, and Dave Thirumalai. "Conformational heterogeneity in human interphase chromosome organization reconciles the FISH and Hi-C paradox." *Nature communications* 10.1 (2019): 1-10) addressed exactly this point raised by the reviewer. In that work, we extracted subpopulations from experimentally measured distributions of distances for CTCF/cohesin loops. Indeed, we found that CTCF/cohesin loops typically have two distinct subpopulations, one for the looped state and one for the unlooped state. This mixture of subpopulations leads to more complex distributions of distances beyond what the present model can capture. This is because the maximum entropy distribution in the present study is a multivariate Gaussian. Multivariate Gaussian is **unimodal** and cannot model multi-modal or mixture systems.

In principle, the DIMES method in the present study can be generalized to model more complex ensembles. For instance, if both the mean distance and mean squared distances are used as constraints, it can be shown that the corresponding maximum entropy distribution can be written as,

$$P(\{\bar{x}_i\}) \propto \exp\left(-\sum_{i<j} k_{ij}(r_{ij} - d_{ij})^2\right)$$

where d_{ij} is “bond” length and k_{ij} is “spring” constant. This distribution, in principle, can model multi-modal or mixture systems. However, the implementation of this model is beyond the scope of the present study, and should definitely be considered in the future. This is an excellent observation.

Comment 5: Figure 4 c is intriguing. However, I am wondering if the author’s model is hiding a limitation regarding triplet interactions. Consider 3 loci: $i > j > k$. If locus i is in contact with locus j (i.e. distance $< 200\text{nm}$) is locus j more likely to be in contact with locus k ? I assume that the authors’ model predicts independence between these two events? However, imaging data and multi-contact sequencing data show hub-like interactions. Can the authors calculate $p(j \text{ contact } k \mid i \text{ contact } j)$ compared to $p(j \text{ contact } k \mid i \text{ not contact } j)$ for both model and imaging data?

Response:

The referee is correct that the event of locus i is in contact with locus j and the event of locus j is in contact with locus k are not independent. If there is a connection between locus i and locus k , then the contact between i and j makes the contact between j and k more likely. Our model **does not predict independence** between these two events. The quantity Z_{ijk} shown in Figure 4c is calculated with respect to its expected value. For an ideal chain, the contact between i and j is independent to the contact between j and k . In essence, Z_{ijk} shows the deviations from the ideal chain, and the independence between the events is captured.

Following the referee's suggestion, we have calculated the following conditional probabilities,

$$Pr(r_{jk} < a | r_{ij} < a)$$

$$Pr(r_{jk} < a | r_{ij} \geq a)$$

We show below the figure for comparison between these two probabilities from imaging data,

Fig. R1 Comparison between $Pr(r_{jk} < a | r_{ij} < a)$ and $Pr(r_{jk} < a | r_{ij} \geq a)$ with $a=300\text{nm}$. Calculated directly from imaging data. Dashed line has slope of 1.

Fig. R1 shows clear dependence between the two events. If they are independent, then the data points should approximately follow the dashed line. Below, Fig. R2 shows the same comparison but computed from the model.

Fig. R2 Comparison between $Pr(r_{jk} < a | r_{ij} < a)$ and $Pr(r_{jk} < a | r_{ij} \geq a)$ with $a=300\text{nm}$. Calculated directly from model. Dashed line has slope of 1.

Fig. R2 shows that our model captures the same dependence as experiment. Furthermore, we show below the direct comparison between experiment and model for $Pr(r_{jk} < a | r_{ij} < a)$ and $Pr(r_{jk} < a | r_{ij} \geq a)$.

Fig. R3 (a) The comparison between experiment and model for $Pr(r_{jk} < a | r_{ij} < a)$ (b) The comparison between experiment and model for $Pr(r_{jk} < a | r_{ij} \geq a)$ Dashed lines have slope of 1.

Fig. R3 shows that our model quantitatively agrees with the imaging data for both $Pr(r_{jk} < a | r_{ij} < a)$ and $Pr(r_{jk} < a | r_{ij} \geq a)$.

Comment 6: The TAD-like pattern section (page 3) seems less thoroughly treated than the other sections. I would suggest either removal or adding in more thorough statistical analysis.

Response:

Following the referee’s suggestion, we choose to perform the calculation of boundary probabilities for Chr21 2 Mbps region. First introduced in Bintu et al 2018 Science, boundary probability is the probability that a genomic locus appears as a single-cell 3D domain boundary. In the revised Fig.5a, we show that our model quantitatively reproduces the boundary probabilities in agreement with the experiment. This shows that the TAD-like domains which are visually observed in the model are also in quantitative agreement with the experiment. We have revised the main text accordingly. On page 4, it reads,

“We then compute the boundary probabilities [48] of individual genomic loci using experimentally measured structures and the structures generated by DIMES. Fig. 5a shows that DIMES is capable to capture the profile of boundary probabilities quantitatively reasonably well, thus establishing suggesting that the DIMES method successfully reproduces the TAD-level structures.”

Comment 7: Can the authors perhaps better separate the biological/physical meaning of k_{ij} - the essential parameters of the model. For instance k_{ij} currently captures both the polymeric nature of chromatin as well as its epigenetic state. Can, for instance, k_{ij} be broken down (i.e. for the chr2 data) into two components: one polymeric component and one epigenetic A/B component i.e.: $k_{ij} \sim |i-j|^c * K_{A/B}$ where c is a scale factor constant and K_A and K_B are two constants independent of i and j ?

Response:

We would like to thank the referee for this insightful suggestion. We have investigated whether k_{ij} can be decomposed into polymer component and epigenetic A/B component. If k_{ij} can be expressed as,

$$k_{ij \in \alpha\beta} \sim f(|i-j|)k_{\alpha\beta}$$

where α and β is the type A or B, and $f(|i-j|)$ is some arbitrary function that only depends on $|i-j|$. $f(|i-j|) \sim |i-j|^c$ being one such example. Then it follows that,

$$\langle k_{ij \in \alpha\beta} \rangle_{|i-j|=s} \sim f(|i-j|) \langle k_{\alpha\beta} \rangle$$

From the equation above, it follows that the **ratio** between $\langle k_{AA}(s) \rangle$, $\langle k_{AB}(s) \rangle$, $\langle k_{BB}(s) \rangle$ should be **independent** of s and be constant. $\langle k_{AA}(s) \rangle$, $\langle k_{AB}(s) \rangle$, $\langle k_{BB}(s) \rangle$ are the quantities that we have calculated and shown in Fig. 7d. Fig. R4 below shows the $\langle k_{BB}(s) \rangle / \langle k_{AA}(s) \rangle$, $\langle k_{BB}(s) \rangle / \langle k_{AB}(s) \rangle$ and $\langle k_{AA}(s) \rangle / \langle k_{AB}(s) \rangle$ as functions of s .

Fig. R4 $\langle k_{BB}(s) \rangle / \langle k_{AA}(s) \rangle$, $\langle k_{BB}(s) \rangle / \langle k_{AB}(s) \rangle$ and $\langle k_{AA}(s) \rangle / \langle k_{AB}(s) \rangle$

Fig. R4 shows that these ratios have large fluctuations and are not independent of s . Hence, we conclude that k_{ij} cannot be easily decomposed as $\langle k_{ij \in \alpha\beta} \rangle_{|i-j|=s} \sim f(|i-j|) \langle k_{\alpha\beta} \rangle$

We then investigate whether if k_{ij} can be decomposed as two additive terms, expressed as,

$$k_{ij \in \alpha\beta} \sim f(|i-j|) + k_{\alpha\beta}$$

It follows that the **difference** between $\langle k_{AA}(s) \rangle$, $\langle k_{AB}(s) \rangle$ and $\langle k_{BB}(s) \rangle$ should be **independent** of s and be constant. Fig. R5 below shows the $\langle k_{AA}(s) \rangle - \langle k_{AB}(s) \rangle$, $\langle k_{AA}(s) \rangle - \langle k_{BB}(s) \rangle$, and $\langle k_{BB}(s) \rangle - \langle k_{AB}(s) \rangle$, as functions of s .

Fig.R5 $\langle k_{AA}(s) \rangle - \langle k_{AB}(s) \rangle$, $\langle k_{AA}(s) \rangle - \langle k_{BB}(s) \rangle$, and $\langle k_{BB}(s) \rangle - \langle k_{AB}(s) \rangle$. Black lines are the moving average over window size of $s=2.6$ Mbps.

Fig. R5 shows that $\langle k_{AA}(s) \rangle - \langle k_{AB}(s) \rangle$, $\langle k_{AA}(s) \rangle - \langle k_{BB}(s) \rangle$, and $\langle k_{BB}(s) \rangle - \langle k_{AB}(s) \rangle$ approximately fluctuating around constant values for $60 \text{ Mbps} < s < 200 \text{ Mbps}$. They show s -dependence for $s < 60 \text{ Mbps}$. We have added a new Figure S7 in the revised SI and revised the main text. On page 7-8 in the discussion section, it now reads,

“Interpretation of k_{ij} ’s: We wondered whether k_{ij} ’s could be decomposed into two additive terms representing the polymer component and epigenetic component, $k_{ij \in \alpha\beta} = k_0(i, j) + k_e(\alpha, \beta)$; $k_0(i, j)$ is the polymer contribution to k_{ij} which only depends on i and j . $k_e(\alpha, \beta)$ is the epigenetic contribution to k_{ij} that only depends on the epigenetic types, e.g. $k_e(A, A)$, $k_e(B, B)$ and $k_e(A, B)$. Assuming that $k_{ij \in \alpha\beta} = k_0(i, j) + k_e(\alpha, \beta)$ holds, by averaging over $s = |i - j|$, it follows that $\langle k_{AA}(s) \rangle - \langle k_{AB}(s) \rangle$, $\langle k_{AA}(s) \rangle - \langle k_{BB}(s) \rangle$, and $\langle k_{BB}(s) \rangle - \langle k_{AB}(s) \rangle$ are independent of s . Supplementary Fig. 7 shows that these quantities approximately fluctuate around constant values for $60 \text{ Mbps} < s < 200 \text{ Mbps}$. This result suggests that, to a first approximation, the parameters k_{ij} ’s may be decoupled into contributions from polymer connectivity and that arising from epigenetic states in an additive manner.”

Reviewer #2 (Remarks to the Author):

This paper reports a maximum entropy-based method (DIMES) to build chromosome structure from distances between chromosome loci measure by imaging techniques. The coefficients (k_{ij}) can be determined by an iterative scaling method. An interesting finding is that the contact triplets in the reconstructed models correlate with CTCF binding sites. Analyzing the TAD

structures and two chromosome compartments of the reconstructed models is useful. Comparing the distance maps derived from Hi-C data with the model built from image data is novel and valuable. Studying how loci deletion may change the structure of chromosomes is also interesting. However, there are some issues that need to be addressed.

Response:

We are very pleased that the referee appreciates our work. We hope that the referee finds our responses to the points raised satisfactory.

(1) The review of the literature is very incomplete. Many methods have been developed for reconstructing genome/chromosome structures from contact maps or distance maps derived from Hi-C data. But few have been reviewed.

Response:

We apologize for not making a proper review of the current literature. The revised manuscript includes additional references. On page 1, it now reads,

“Data-driven approaches [29-38] have been advanced to solve the complicated Hi-C to 3D structure problem (see the summary in Ref. [40] for additional related studies and Ref. [41] for a comprehensive overview of the existing methods).”

We have added the following references:

1. Z. Duan, M. Andronescu, K. Schutz, S. McIlwain, Y. J. Kim, C. Lee, J. Shendure, S. Fields, C. A. Blau, and W. S. Noble, A Three-Dimensional Model of the Yeast Genome, *Nature (London)* 465, 363 (2010).
2. R. Kalhor, H. Tjong, N. Jayathilaka, F. Alber, and L. Chen, Genome Architectures Revealed by Tethered Chromosome Conformation Capture and Population-Based Modeling, *Nat. Biotechnol.* 30, 90 (2012).
3. M. Rousseau, J. Fraser, M. A. Ferraiuolo, J. Dostie, and M. Blanchette, Three-Dimensional Modeling of Chromatin Structure from Interaction Frequency Data Using Markov Chain Monte Carlo Sampling, *BMC Bioinf.* 12, 414 (2011).
4. Z. Zhang, G. Li, K.-C. Toh, and W.-K. Sung, 3D Chromosome Modeling with Semi-Definite Programming and Hi-D Data, *J. Comput. Biol.* 20, 831 (2013).
5. M. Hu, K. Deng, Z. Qin, J. Dixon, S. Selvaraj, J. Fang, B. Ren, J. S. Liu, and A. Tanay, Bayesian Inference of Spatial Organizations of Chromosomes, *PLoS Comput. Biol.* 9, e1002893 (2013).
6. N. Varoquaux, F. Ay, W. S. Noble, and J.-P. Vert, A Statistical Approach for Inferring the 3D Structure of the Genome, *Bioinformatics* 30, i26 (2014).
7. Oluwadare, Oluwatosin, Yuxiang Zhang, and Jianlin Cheng. "A maximum likelihood algorithm for reconstructing 3D structures of human chromosomes from chromosomal contact data." *BMC genomics* 19.1 (2018): 1-17.

8. Wang, Hao, et al. "Reconstruct high-resolution 3D genome structures for diverse cell-types using FLAMINGO." *Nature Communications* 13.1 (2022): 1-18.
9. Oluwadare, Oluwatosin, Max Highsmith, and Jianlin Cheng. "An overview of methods for reconstructing 3-D chromosome and genome structures from Hi-C data." *Biological procedures online* 21.1 (2019): 1-20.
10. N. Hua et al., Producing Genome Structure Populations with the Dynamic and Automated PGS Software, *Nat. Protoc.* 13, 915 (2018).
11. Hua, Nan, et al. "Producing genome structure populations with the dynamic and automated PGS software." *Nature protocols* 13.5 (2018): 915-926.

(2) The maximum entropy approach has some similarity with the maximum likelihood approach for reconstructing chromosome structures from Hi-C data that is based on Gaussian distribution on the distances between chromosome loci in the literature. But the connection is not drawn in the paper.

Response:

We can only guess that the reviewer is referring to *Oluwadare, Oluwatosin, Yuxiang Zhang, and Jianlin Cheng. "A maximum likelihood algorithm for reconstructing 3D structures of human chromosomes from chromosomal contact data." BMC genomics 19.1 (2018): 1-17.*

We would like to thank the referee for pointing out this work (if this work is indeed what the referee has in mind), which we were unaware of when submitting our manuscript. In this work, the pairwise distance between two loci is assumed to be Gaussian,

$$P(r_{ij}) \sim \frac{1}{\sigma\sqrt{2\pi}} \exp\left(-\frac{1}{2\sigma^2}(r_{ij} - r_{ij}^0)^2\right)$$

where r_{ij}^0 is the inferred distance converted from Hi-C data by a power law. The distribution of 3D structure (coordinates) is given below, under the assumption that pairwise distances are independent of each other,

$$P(\{\vec{x}\}) \sim \prod_{i,j} \frac{1}{\sigma\sqrt{2\pi}} \exp\left(-\frac{1}{2\sigma^2}(r_{ij} - r_{ij}^0)^2\right)$$

Then a single 3D structure is obtained by finding the coordinates that maximize the log-likelihood,

$$\log \prod_{i,j} \frac{1}{\sigma\sqrt{2\pi}} \exp\left(-\frac{1}{2\sigma^2}(r_{ij} - r_{ij}^0)^2\right)$$

First, the essences of the two models both are to construct the distribution of 3D structures constrained by the experiment data and find the structures accordingly. The 3D structure obtained by the above maximum likelihood method and the **average** structure obtained from our model would likely agree to some degree. We also note that if σ is taken to be a control parameter, σ_{ij} and r_{ij}^0 are taken to be zero. Then the resulting distribution of coordinates is as same as Eq. 1 in the main text.

We would also like to point out two key differences: 1. the maximum likelihood method described above obtains a single “best fit” or “consensus” structure, whereas our model generates an ensemble, which corresponds to reality as cell-to-cell variations in the structures show. 2. The distribution of distances in principle cannot be Gaussian since distances are always non-negative. 3. The underlying basis for the two methods are different: the maximum likelihood method seeks the structure that maximizes the likelihood of a pre-defined distribution, whereas the maximum entropy principle is to find the distribution that assumes no more than the data suggests (e.g. maximum entropy).

Based on the reasons mentioned, we are unsure whether a rigorous connection between our maximum entropy method and the maximum likelihood method in *Oluwadare et al BMC genomics 2018* can be easily established.

(3) In the results section, the validation of the method is based on how well the reconstructed model fits all the experimental distance data, which is not sufficiently rigorous and can easily overfit the data. A more rigorous cross-validation procedure should be used. That is, use a portion of distance data to build the chromosome model and then use the remaining distance data to validate the model.

Response:

We would like to thank the referee for this valuable suggestion. Following the suggestion, we have performed the cross-validation. In the revised SI, we added a new SI Figure 9. In summary, the results show that the model is remarkably stable for the missing data. Even when only 10% of pairwise distances are used to build the model, the rest 90% of the distances can still be predicted quantitatively remarkably well. This also implies that the model does not overfit.

On page 2 in the revised main text, it now reads,

“We then perform cross-validation of the DIMES method. This is done as follows: a fraction of pairwise distances is randomly chosen from the distance map and is deemed to be missing data. The new distance map containing missing data is then used as input for DIMES. It is important to note that the missing data is not used to update k_{ij} . Finally, the predicted distances for the missing data are compared with the values obtained from the full distance map. Fig. 2(c-f) compares the input distance maps with missing data and the full predicted distance maps. Remarkably, DIMES quantitatively predicts pairwise distances for the missing data even if only 10% of the distance map is used. Together, these results demonstrate that the model is effective in producing 3D structures that are consistent with the experimental input and is robust with respect to missing data.”

(4) There is no comparison between DIMES and existing distance based methods of reconstructing 3D chromosome structures from Hi-C data that can also be used to reconstruct 3D chromosome structures from image data in this work.

Response:

To the best of our knowledge, in most if not all the literature, the accuracy of a method is mainly evaluated by its capability of recapitulating the contact probability or inferred pairwise distances between genomic loci from Hi-C. Typically, these models use functions with unknown parameters that are evaluated by fitting to the Hi-C data. Spearman or Pearson correlation are widely used metrics to quantify such accuracy. Our model, based on the maximum entropy principle, theoretically guarantees that the pairwise distances are **accurately** reproduced. We show in Fig.2 in the main text that the model can accurately reproduce the input target. The Pearson correlation coefficient is near unit (> 0.99). If we use how accurately the model reproduces the input data as a metric, then our model has an accuracy of almost 100%. Technically, unlike other methods where experiment data is used as a target to “fit”, our model uses experimental data as **direct** constraints. Hence, we believe our model cannot be properly compared with other data-driven methods or polymer models easily.

In addition, the main focus of the present study is to show that our model, with high accuracy, can reproduce the **entire ensemble of 3D structures, including pairwise distances and beyond**. We validate the model **by directly comparing the prediction with the imaging data**. To the best of our knowledge, we are not aware of other works that conduct similar investigations. The focus of the present study is not to benchmark the performance of our method and compare it with others. And we would argue that there are no parallel methods in the literature we can readily compare to.

Furthermore, we agree with the referee that some of the existing methods in the literature can be modified to apply to the imaging data. However, these modifications are not explored by the original authors or other authors. We trust that the Referee agrees with our explanation, and understands that comparison with a other methods, including the ones we introduced, is overly burdensome. It could be a great idea for a technical review of the methods.

Reviewer #3 (Remarks to the Author):

The authors report a new computational method for modeling ensembles of 3D chromatin structure that relies on pairwise distances between genomic loci derived from direct imaging experiments. The authors specifically apply this new method, DIMES, to results recently reported by Bintu et al (Science 2018, super resolution of 2 Mb region on chr21) and Su et al (Cell 2020, massive multiplex FISH over 23 chromosomes).

Response:

We thank the reviewer for the summary of our work.

Structural ensembles predicted by DIMES, using the maximum entropy principle, are randomly sampled, deriving predicted spatial distances the authors then compare with target distance

map measurements. The authors first validate the accuracy of the models predicted by DIMES by comparison with the experimental input. Can the authors elaborate or further clarify the significance of their reported validation results? For example, the authors claim that using squared distances yield better numerical convergence, but I cannot find a comparison of alternative models tested (e.g. average distance).

Response:

We should have used “computation cost” instead of “convergence” . As shown in the Eq. 1 in the main text, if average **squared** distances are used as constraints, the maximum entropy distribution is multivariate normal. The advantage of multivariate normal is that it allows direct and efficient sampling and direct calculation of $\langle r_{ij}^2 \rangle$ (see the text below Eq. 16 in the SI).

On the other hand, if average distances are used as constraints, the maximum entropy distribution is given by,

$$P(\{\bar{x}_i\}) \propto \exp\left(-\sum_{i < j} k_{ij} \left| |\bar{x}_i - \bar{x}_j| \right| \right)$$

To the best of our knowledge, there is no direct sampling method for this distribution. Markov chain Monte-Carlo or other similar methods are needed to sample this distribution. Hence, we argue that choosing average squared distances as constraints yields minimal computation cost.

We have revised the main text accordingly. On page 2 in the revised main text, it now reads,

“However, constraining average squared distances is computationally more efficient because the resulted maximum entropy distribution is a multivariate Gaussian distribution which allows fast sampling.”

In this report, the authors use DIMES to make a number of relevant observations related to chromosome organization, predicting the distribution of pair-wise distances in structural ensembles, and recovering higher-order structures ranging from three-way contacts and TAD-like patterns to the overall size and shape of chromosome organization. When applied to chromosome-scale imaging data (Su et al.), the authors again recover higher-order chromatin structures, including A/B compartments, and further demonstrate significant variation among DIMES ensemble structures for Chr2. Can the authors comment on the specific experimental data that supports this level of variation (noted in Supp Fig 2), as well as elaborate on the level to which gaps between imaging loci would potentially contribute to these observations?

Response:

We thank the referee for this interesting question. In the new SI Fig.8, we show two examples of single-cell distance maps for Chr2 using imaging data from Su et al. They show that level of

variation observed in the model is also present in the imaging data. We further compare the distributions of distances between DIMES and the imaging data for two different pairs. The result shows that our model also captures the degree of variation quantitatively. On page 4 in the revised main text, section “Structures of the 242Mbps-long Chr2”, it now reads,

“Experimental single-cell distance maps (Supplementary Fig. 8a) show that similar variations are also observed in vivo. DIMES reproduces the distributions of pairwise distances at large length scales (Supplementary Fig. 8b).”

We believe that the gaps between imaging loci will not affect our conclusions. At this juncture, we can only conclude that if higher level resolution in the imaging data is available, it can only fortify our conclusions by providing higher density of points. Of course, assessing this awaits future experiments.

Beyond the development and applications of DIMES, the results presented offer some valuable insight into genome organization. Ensemble structures for chromosome-scale distances suggest that the levels of A-A and B-B compartment genomic distance preferences (inferred as interaction strengths) are different between chromosome 2 and chromosome 21, as well as between differing length scales. This is an interesting finding, and I’m wondering if the authors can speculate on potential mechanisms.

Response:

First, we thank the reviewer for the compliments. We would like to address the referee’s comments by two points:

1. The A-A and B-B interaction strengths are different between different length scales. In the revised version of the main text, we have attempted to decompose the A-A and B-B interactions into one polymeric component and one epigenetic component. We found that the polymeric contribution to the interaction strength is length-scale dependent, as expected for polymers. The epigenetic component is approximately length-scale independent. This is consistent with the underlying molecular mechanism that governs preferential A-A and B-B interaction. If we assume that the preferential A-A and B-B interactions come from the molecular interaction mediated by various epigenetic markers (histone modifications etc), then such interactions should **not** be genomic distance dependent.
2. The A-A and B-B interaction strengths differ between chromosome 2 and chromosome 21. We speculate that the potential origin of such difference is that the average epigenetic profile for A and B compartments is different between Chr2 and Chr21. In addition, A-A and B-B interaction strength also has polymer contribution. And Chr2 and Chr21 may have different polymeric characteristics which further leads to the difference in A-A and B-B interaction strengths.

The authors further demonstrate the value of DIMES by applying their methods to examples of genomic structural variants (structures predicted by DIMES are highly consistent with experimental observations). The authors conclude with a comparison of Hi-C and imaging data and identify a number of interesting divergences between these approaches, specifically in the extent of long-range interactions and A-A and B-B compartment interaction strengths.

The developed method and reported findings are of value to the field and are worthy of publication. In general, the manuscript would be improved with an extended discussion of these findings.

Response:

We hope that the referee is satisfied with our responses to the comments above. We are very pleased that this reviewer appreciates our work, and states that the work is suitable for publication in this journal.

REVIEWERS' COMMENTS

Reviewer #2 (Remarks to the Author):

The authors addressed my review comments well. The method is innovative and the result are convincing. The manuscript has been improved after the revision.

Reviewer #3 (Remarks to the Author):

The authors addressed the points I made in the initial review. The manuscript should be published.

One minor point: chromatins (as written in the revision title) should read chromatin

Responses to Reviewers:

Reviewer #2 (Remarks to the Author):

The authors addressed my review comments well. The method is innovative and the result are convincing. The manuscript has been improved after the revision.

Response:

We are very glad that the reviewer is satisfied with our response and he/her finds our work innovative and convincing.

Reviewer #3 (Remarks to the Author):

The authors addressed the points I made in the initial review. The manuscript should be published.

One minor point: chromatins (as written in the revision title) should read chromatin

Response:

We are glad that the reviewer finds our response satisfactory. We have corrected the “chromatins” to “chromatin” in the title.